# Cost-aware LLM-based Online Dataset Annotation

**Eray Can Elumar** [*]
Dept. of Electrical and Computer Eng.
Carnegie Mellon University
Pittsburgh, PA 15213
eelumar@andrew.cmu.edu

**Cem Tekin**
Dept. of Electrical and Electronics Eng.
Bilkent University
Ankara, Türkiye
cemtekin@ee.bilkent.edu.tr

**Osman Yağan**
Dept. of Electrical and Computer Eng.
Carnegie Mellon University
Pittsburgh, PA 15213
oyagan@ece.cmu.edu

## Abstract

Recent advances in large language models (LLMs) have enabled automated dataset labeling with minimal human supervision. While majority voting across multiple LLMs can improve label reliability by mitigating individual model biases, it incurs high computational costs due to repeated querying. In this work, we propose a novel online framework, Cost-aware Majority Voting (CaMVo), for efficient and accurate LLM-based dataset annotation. CaMVo adaptively selects a subset of LLMs for each data instance based on contextual embeddings, balancing confidence and cost without requiring pre-training or ground-truth labels. Leveraging a LinUCB-based selection mechanism and a Bayesian estimator over confidence scores, CaMVo estimates a lower bound on labeling accuracy for each LLM and aggregates responses through weighted majority voting. Our empirical evaluation on the MMLU and IMDB Movie Review datasets demonstrates that CaMVo achieves comparable or superior accuracy to full majority voting while significantly reducing labeling costs. This establishes CaMVo as a practical and robust solution for cost-efficient annotation in dynamic labeling environments.

## 1 Introduction

The rapid proliferation of data across domains has created an urgent need for accurate, large-scale annotation pipelines. While human experts and crowd workers have been the gold standard for dataset labeling, manual annotation is notoriously slow, expensive, and prone to inter-annotator inconsistency Petrović et al. [2020]. As machine learning models become increasingly sophisticated, their demand for high-quality, richly labeled datasets only intensifies, exacerbating this bottleneck.

Recent advances in large language models (LLMs) offer a promising remedy: by leveraging transformer-based architectures such as GPT, it is now possible to automate much of the labeling workload. LLMs excel at natural-language understanding, reasoning, and contextual inference, enabling rapid generation of annotations with minimal human effort Naveed et al. [2023].

However, relying on a single LLM introduces issues of biases inherited from its training data and stochastic variability across repeated queries, undermining reliability and reproducibility Errica et al. [2024], Li et al. [2024a]. A common strategy to bolster label quality is ensembling: querying multiple LLMs, or multiple samples from the same model, and aggregating their outputs via majority voting.

---

[*]Work done while at Carnegie Mellon University. Now at Intel Corporation.

39th Conference on Neural Information Processing Systems (NeurIPS 2025).

This reduces hallucinations and offsets the bias of individual models but substantially increases cost, as each additional model increases latency and compute expenditure Yang et al. [2023]. In practice, querying every available LLM for each instance is often wasteful and unnecessary.

In this paper, we address this trade-off by *adaptively selecting a subset* of LLMs for majority voting on each input, achieving comparable accuracy to full-ensemble voting while dramatically cutting cost. Unlike prior work on LLM weight optimization or query routing—which presumes access to ground-truth labels or a pre-trained routing model Chen et al. [2024], Nguyen et al. [2024], Ding et al. [2024]—our method operates *online*, without any held-out training set or ground truth.

Our contributions are as follows:

1. **Online formulation.** To the best of our knowledge, this is the first work on LLM-based dataset labeling in which both vote weights and the queried subset of LLMs are adapted in real time, i.e. without relying on a pre-trained model or a dedicated training set.

2. **Cost-aware Majority Voting (CaMVo).** We propose CaMVo, an algorithm that combines a LinUCB-style contextual bandit with a Bayesian Beta-mixture confidence estimator. For each candidate LLM, CaMVo computes a lower confidence bound on its correctness probability given the input's embedding, then selects the smallest-cost subset whose aggregated confidence exceeds a user-specified threshold $\delta$.

3. **Empirical validation.** Through experiments on the MMLU benchmark and the IMDB Movie Review dataset, we show that CaMVo matches or exceeds the accuracy of full majority voting (majority voting with all available LLMs) while significantly reducing the cost: on MMLU, CaMVo achieves higher accuracy with around $40\%$ lower cost; on IMDB, it attains only $0.17\%$ drop in accuracy while halving query expenditure.

## 1.1 Related Work

**Ensembling and Majority Voting with LLMs.** Aggregating outputs from multiple LLMs (or repeated queries to a single LLM) via majority voting has become a popular strategy to boost annotation reliability. Chen et al. [2024] analyze the effect of repeated queries to a single model and observe a non-monotonic accuracy curve: performance improves initially but degrades beyond an optimal number of calls due to task heterogeneity. While additional LM calls enhance accuracy on easier queries, they may introduce noise or inconsistency that degrades performance on more challenging ones. To address this, the authors propose a scaling model that predicts the optimal number of LM calls required to maximize aggregate performance. Trad and Chehab [2024] compare re-querying and multi-model ensembles, showing that ensemble strategies are most effective when individual models or prompts exhibit comparable performance levels, and ensemble gains may diminish when individual model accuracies diverge.

Yang et al. [2023] propose a weighted majority-voting ensemble for medical QA, combining dynamic weight adjustment with clustering-based model selection. However, their approach relies on offline training data and focuses solely on improving accuracy, without accounting for the cost of querying. In contrast, our approach selects a cost-effective subset of heterogeneous LLMs online, without any pre-training.

**LLM Query Routing.** Query routing addresses the problem of selecting a single LLM per query to optimize cost or latency under performance constraints. Nguyen et al. [2024] cast LLM selection as a contextual bandit problem, training offline on labeled data to learn a routing policy that maps query embeddings to a single optimal LLM under a total budget constraint of $b$. Ding et al. [2024] train a router to distinguish "easy" versus "hard" queries, sending easy tasks to local LLMs and hard ones to cloud APIs. Unlike these methods, our method operates in a fully online setting without access to labeled training data, updating the model dynamically during the labeling process. Moreover, rather than routing to a single LLM, we select a cost-efficient subset for majority voting.

**Confidence Estimation in LLM Outputs.** Estimating model confidence can guide automatic label selection. Kadavath et al. [2022] explore how a language model's own uncertainty estimates can serve as predictors of answer correctness, and find that the cumulative log-probability the model assigns to its generated token sequence correlates strongly with factual accuracy across diverse benchmarks. Li et al. [2024a] generate code multiple times and use output similarity as a proxy for confidence,

choosing the most consistent result. While these works focus on self-consistency of a single model, we estimate a probabilistic lower bound on each LLM's correctness via a Bayesian Beta-mixture model, incorporating both past performance and context.

**Weighted Majority Voting.** Beyond simple voting, weighted schemes assign each annotator or model a reliability score such as the accuracy of the annotator, or the label confidence reported by the annotator. One notable approach is the GLAD model by Whitehill et al. [2009], which formulates weighted majority voting as a probabilistic inference problem over annotator expertise and task difficulty. GLAD jointly estimates per-annotator reliability and per-item ambiguity via a generative model, using an EM algorithm to infer the latent variables. Li and Yu [2014] introduce Iterative Weighted Majority Voting (IWMV) to aggregate noisy crowd labels by iteratively estimating worker reliability, and show it approaches the oracle Maximum A Posteriori (MAP) solution.

**Crowdsourcing.** Rangi and Franceschetti [2018] utilize the bandits-with-knapsacks framework to dynamically estimate worker accuracy and allocate tasks in real time to maximize overall labeling quality within a budget. Another influential model is by Raykar et al. [2010], which jointly infers true labels and annotator reliabilities by modeling each worker's confusion matrix. Through an expectation–maximization procedure, the method down-weights inconsistent annotators and yields more accurate aggregated labels without prior knowledge of worker quality. Our method parallels this online estimation but differs in that it leverages contextual embeddings and targets LLM ensembles rather than human annotators.

The comparison of our work with prior work is summarized in Table 1

| Method | Ensemble Type | Pretrained | Online | Contextual |
|---|---|---|---|---|
| Ours (CaMVo) | Subset voting | No | Yes | Yes |
| Yang et al. [2023] | Weighted voting | Yes | No | Yes |
| Nguyen et al. [2024] | Single-model routing | Yes | No | Yes |
| Ding et al. [2024] | Single-model routing | Yes | No | Yes |
| Chen et al. [2024] | Re-querying | No | No | No |
| Li et al. [2024a] | Re-querying | No | No | Yes |
| Li and Yu [2014] | Crowd aggregation | No | Yes | No |
| Rangi and Franceschetti [2018] | Crowd assignment | No | Yes | No |
| Raykar et al. [2010] | Crowd aggregation | No | Yes | No |

Table 1: Comparison of our work with prior ensemble and routing approaches.

## 2 Problem Statement

In this section, we formally define the problem setting, introduce the baseline algorithm, and provide some results that will be used by our proposed algorithm, which will be introduced in §3.

Consider an unlabeled dataset $\mathcal{D} = \{x_1, x_2, \ldots, x_T\}$, where each $x_t$ denotes a data instance (e.g., a text sample). Let there be a set $[K]$ of $K$ distinct large language models (LLMs), where each LLM $l_i$ is associated with a known cost per token $\rho_i$ and can be represented as a function $l_i : \mathcal{Q} \to \mathcal{R}$, mapping a query $q \in \mathcal{Q}$ to a response $r \in \mathcal{R}$. We denote the total number of possible labels for the dataset $\mathcal{D}$ as $M$. The objective in this setting is to assign a predicted label $\hat{y}_t \in [M]$ to each data instance $x_t$ by querying a subset of the available LLMs and aggregating their outputs. Labeling is performed sequentially, where each data instance $x_t$ is processed in round $t$. In each round, LLMs are queried independently of other LLMs and without memory of prior interactions (i.e., no context is preserved between rounds). Furthermore, all queries are made in a zero-shot setting, meaning that no task-specific fine-tuning or additional training data is used.

To serve as a baseline, we introduce the following weighted majority voting scheme. In this scheme, the predicted label $\hat{y}_t$ for instance $x_t$ is determined by aggregating the votes of all $K$ LLMs using:

$$\hat{y}_t = \arg \max_{m \in [M]} \sum_{i=1}^{K} \omega_{\text{def},i}(t) \cdot \mathbb{1}\{y_{i,t} = m\}, \tag{1}$$

where $\mathbb{1}\{\cdot\}$ is the indicator function that returns 1 if the condition is true and 0 otherwise, and $y_{i,t}$ denotes the label outputted by LLM $l_i$ for instance $x_t$. Since the true label is not available in our setting, we use the empirical accuracy of model $l_i$ relative to the predicted labels as the voting weight. Hence, $\omega_{\text{def},i}(t) = (\sum_{s=1}^{t-1} \mathbb{I}(y_{i,s} = \hat{y}_s))/N_{i,t}$, where $N_{i,t}$ denotes the number of times LLM $l_i$ has been queried up to round $t$, and $\hat{y}_s$ is the predicted label for round $s$. The pseudocode for this scheme, which we refer to as the *baseline method*, is provided in Algorithm 2 in Appendix C.

The goal is to label the dataset $\mathcal{D}$ in a cost-efficient manner by dynamically selecting a subset of LLMs for each data instance $x_t$. To facilitate this selection, we assume access to a model $\text{Emb}(\cdot)$ that generates a $d$-dimensional embedding $\mathbf{e}_t = \text{Emb}(x_t)$ for $x_t$ in round $t$. This embedding serves as a representation of the instance and plays an analogous role to the *context* in a contextual bandit framework. Using this embedding, our approach, which will be introduced in § 3, estimates a lower bound on the probability that a given LLM $l_i$ will produce the correct label for $x_t$, and utilizes this bound to determine the subset of LLMs. The details for estimating this lower bound is given in § 3.

Naturally, selecting only a subset of LLMs rather than querying all available models may result in reduced labeling accuracy. To manage this trade-off, we introduce a user-defined parameter $\delta \in [0, 1]$ that specifies the desired minimum relative confidence of the selected subset compared to the full majority vote using all $K$ LLMs. Let $L_{i,t}$ denote the lower confidence bound on the estimated probability that LLM $l_i$ correctly labels instance $x_t$ at round $t$. Our algorithm identifies the cost-minimizing subset of LLMs whose aggregated confidence, relative to that of the baseline method, satisfies the accuracy constraint imposed by $\delta$.

We will leverage the following result to estimate the confidence of the majority vote label based on the $L_{i,t}$ and $\omega_{i,t}$ values when majority voting is performed over a subset $\mathcal{A}$ of LLMs.

**Lemma 2.1.** *Let $\omega_i$ denote the weight and $L_i$ the lower confidence bound on the correctness of the output from LLM $l_i$. Suppose the outputs of LLMs are conditionally independent given the data instance. Then, for a subset of LLMs $\mathcal{A} \subseteq [K]$, the lower bound on the probability that majority voting over the subset yields the correct label is given by*

$$\delta_{\mathcal{A}}(\boldsymbol{L}, \boldsymbol{\omega}) = \sum_{\substack{S \subseteq \mathcal{A} \\ \sum_{r \in S} \omega_r > \frac{W_{\mathcal{A}}}{2}}} \prod_{i \in S} L_i \prod_{j \in \mathcal{A} \setminus S} (1 - L_j), \tag{2}$$

*where $W_{\mathcal{A}} = \sum_{i \in \mathcal{A}} \omega_i$. Proof of this result is provided in Appendix A.*

For practical purposes and computational efficiency, this expression may be approximated as $\delta_{\mathcal{A}}(\boldsymbol{L}, \boldsymbol{\omega}) \approx 1 - F_{\text{Beta}}(0.5; W_{L,\mathcal{A}}, W_{\mathcal{A}} - W_{L,\mathcal{A}})$, where $F_{\text{Beta}}(x; \alpha, \beta)$ is the CDF of a $\text{Beta}(\alpha, \beta)$ distribution evaluated at point $x$, $W_{L,\mathcal{A}} = \sum_{i \in \mathcal{A}} \omega_i \cdot L_i$, and $W_{\mathcal{A}} = \sum_{i \in \mathcal{A}} \omega_i$.

Note that Lemma 2.1 can be extended to remove the conditional-independence assumption using Bahadur's model, which characterizes the joint distribution of binary variables using their marginal probabilities, second-order dependence terms known as Bahadur parameters, and higher-order interaction terms [Bahadur, 1961]. This framework allows the computation of majority vote confidence by summing the probabilities of all outcomes where the majority label is correct. However, the absence of a closed-form expression and the exponential growth in computational complexity with the number of variables limit its practical use. In practice, Monte Carlo–based simulation methods can approximate the majority-vote confidence by generating correlated LLM outputs; we describe and analyze CaMVo with one such approach in §G.

Finally, we introduce a user-specified parameter $k_{\min}$, which enforces a floor on the number of LLMs queried per instance. By requiring at least $k_{\min}$ votes in every round, this constraint further safeguards label quality—ensuring that no annotation is based on fewer than $k_{\min}$ model predictions.

The proposed Cost-aware Majority Voting (CaMVo) algorithm, which incorporates these results and user-defined parameters, is presented in §3.

## 3 The CaMVo Algorithm

We propose a novel algorithm, *Cost-aware Majority Voting* (CaMVo), for efficient dataset labeling with large language models (LLMs). CaMVo aims to select a cost-effective subset of LLMs for each input instance by leveraging contextual embeddings to estimate a lower confidence bound on

the probability that each model will produce a correct label. These bounds are computed using a LinUCB-based framework [Li et al., 2010]. Based on this information, CaMVo identifies a subset of LLMs such that the confidence of their weighted majority vote exceeds a user-specified threshold $\delta$. If no such subset exists, the algorithm defaults to querying all available models. The pseudo-code of CaMVo is provided in Algorithm 1, and consists of six main steps described below.

---

**Algorithm 1** Cost-aware Majority Voting (CaMVo) Algorithm

---

1: **Input:** Set of LLMs $[K]$, cost per token $\rho_i$ for each LLM $i$, embedding model $\text{Emb}(\cdot)$, confidence threshold $\delta$, LinUCB regularization parameter $\lambda_L$, exploration parameter $\alpha$, regularization parameter $\lambda_R$
2: $A_{i,0} \leftarrow \lambda_L I_d$, $\boldsymbol{b}_{i,0} \leftarrow \boldsymbol{0}_d$, $\forall i \in [K]$
3: **for** each round $t = 1, 2, \ldots, T$ **do**
4:     Get context vector: $\boldsymbol{e}_t \leftarrow \text{Emb}(x_t)$
5:     **for** each LLM $i = 1, 2, \ldots, K$ **do**
6:         $q_{i,t}(\boldsymbol{e}_t) \leftarrow \boldsymbol{e}_t^\top A_{i,t-1}^{-1} \boldsymbol{b}_{i,t-1}$
7:         $\theta_{i,t}(\boldsymbol{e}_t) = q_{i,t}(\boldsymbol{e}_t) - \alpha\sqrt{\boldsymbol{e}_t^\top A_{i,t-1}^{-1} \boldsymbol{e}_t}$
8:         $\bar{L}_{i,t} \leftarrow \text{Est}_i(\theta_{i,t}(\boldsymbol{e}_t))$
9:         $L_{i,t} \leftarrow \frac{\bar{L}_{i,t} \cdot N_{i,t-1} + \lambda_R \cdot \log(t+1)/2}{N_{i,t-1} + \lambda_R \cdot \log(t+1)}$
10:        $\omega_{i,t} \leftarrow \mu_{i,t-1} \cdot q_{i,t}(\boldsymbol{e}_t)$
11:     **end for**
12:     $\mathcal{A}_t \leftarrow \text{Oracle}(\mathbf{L}_t, \boldsymbol{\omega}_t, \delta, k_{\min})$
13:     Query LLMs: $y_{i,t} = l_i(x_t)$, $i \in \mathcal{A}_t$
14:     $\hat{y}_t \leftarrow \arg\max_m \sum_{i \in \mathcal{A}_t, y_{i,t}=m} \omega_i(t)$
15:     **if** $|\mathcal{A}_t| > 1$ **then**
16:         **for** each LLM $i \in \mathcal{A}_t$ **do**
17:             $r_{i,t} \leftarrow \mathbb{1}\{y_{i,t} = \hat{y}_t\}$
18:             Update: $N_{i,t} \leftarrow N_{i,t-1} + 1$
19:             Update: $A_{i,t} \leftarrow A_{i,t-1} + \boldsymbol{e}_t \boldsymbol{e}_t^\top$
20:             Update: $\boldsymbol{b}_{i,t} \leftarrow \boldsymbol{b}_{i,t-1} + r_{i,t}\boldsymbol{e}_t$
21:             Update: $\mu_{i,t} \leftarrow \frac{\sum_{s=1}^t \mathbb{1}\{y_{i,s}=\hat{y}_s\}}{N_{i,t}}$
22:             Update the parameters of $\text{Est}_i$
23:         **end for**
24:     **end if**
25: **end for**

---

**LinUCB-Based Confidence Estimation.** For each LLM $l_i$, CaMVo maintains a matrix $A_i \in \mathbb{R}^{d \times d}$ and vector $\boldsymbol{b}_i \in \mathbb{R}^d$, initialized as $A_{i,0} = \lambda_L I_d$, $\boldsymbol{b}_{i,0} = \boldsymbol{0}$, where $\lambda_L > 0$ is a user-defined parameter. Given $\boldsymbol{e}_t = \text{Emb}(x_t)$, the estimated confidence, and its confidence bound is computed as:

$$q_{i,t}(\boldsymbol{e}_t) = \boldsymbol{e}_t^\top A_{i,t-1}^{-1} \boldsymbol{b}_{i,t-1}, \quad C_{i,t}(\boldsymbol{e}_t) = \alpha\sqrt{\boldsymbol{e}_t^\top A_{i,t-1}^{-1} \boldsymbol{e}_t},$$

From these, the lower confidence bound (LCB) of LLM confidence can be found as:

$$\theta_{i,t}(\boldsymbol{e}_t) = q_{i,t}(\boldsymbol{e}_t) - C_{i,t}(\boldsymbol{e}_t).$$

Since this is an LCB of a probability, we clip it between 0 and 1 when it is out of range [0,1].

**Bayesian Estimation of Label Correctness.** Given the inherent probabilistic nature of LLM outputs, the estimated confidence score may not reliably indicate the correctness of a label. To address this, we introduce a Bayesian estimator $\text{Est}_i(\cdot)$ that models the posterior probability that $l_i$'s prediction is correct, conditioned on this confidence. First, we define a latent variable $h_{i,t} = \mathbb{1}\{y_{i,t} = \hat{y}_t\}$, where $\hat{y}_t$ is the assigned label. Note that ideally, the true label should be used instead of $\hat{y}_t$, but since we do not have access to the true label, we instead use $\hat{y}_t$. We model the conditional likelihood of $q_{i,t}(\boldsymbol{e}_t)$, given the latent variable $h_{i,t}$, as a Beta-distributed random variable:

$$q_{i,t}(\boldsymbol{e}_t) \mid h_{i,t} = 1 \sim \text{Beta}(\alpha_{i,1}, \beta_{i,1}), \quad q_{i,t}(\boldsymbol{e}_t) \mid h_{i,t} = 0 \sim \text{Beta}(\alpha_{i,0}, \beta_{i,0}).$$

Further, to model $\mathbb{P}(h_{i,t} = 1)$, we use the empirical historical relative accuracy of $l_i$, $\mu_{i,t-1}$, which captures the accuracy of LLM $l_i$ relative to the predicted labels up to round $t - 1$. Similarly, $\mathbb{P}(h_{i,t} = 0) = 1 - \mu_{i,t-1}$. Applying Bayes' rule with these models, the posterior probability is modeled as:

$$\text{Est}_i(q) = \mathbb{P}(h_{i,t} = 1 \mid q) = \frac{\mu_{i,t-1} \cdot \text{Beta}_i(q; \alpha_{i,1}, \beta_{i,1})}{\mu_{i,t-1} \cdot \text{Beta}_i(q; \alpha_{i,1}, \beta_{i,1}) + (1 - \mu_{i,t-1}) \cdot \text{Beta}_i(q; \alpha_{i,0}, \beta_{i,0})},$$

We apply this estimator to $\theta_{i,t}(\boldsymbol{e}_t)$, the LCB of the estimated LLM confidence as $\bar{L}_{i,t} = \text{Est}_i(\theta_{i,t}(\boldsymbol{e}_t))$ to encourage exploration under the UCB principle. Note that unlike traditional UCB-based methods that promote exploration via upper bounds, we use the LCB as it expands the size of the set of LLMs likely to satisfy the confidence threshold $\delta$.

**Regularization.**    In the absence of ground-truth labels, empirical relative accuracy estimates in majority voting can overfit, resulting in overconfident weights and biased aggregation. Further, since subset selection is based on these estimated confidences, early estimation errors can bias the selection process and lead to compounding errors over time. To address this issue, we regularize the LCB of the estimated confidence of LLM $l_i$ using Laplace smoothing:

$$L_{i,t} = \frac{\bar{L}_{i,t} \cdot N_{i,t} + \lambda_R \cdot \log(t+1)/2}{N_{i,t} + \lambda_R \cdot \log(t+1)} \tag{3}$$

where $\lambda_R > 0$ is a user-defined regularization parameter that controls the strength of smoothing. Laplace smoothing is chosen since it corresponds to using a uniform prior on the Beta estimator, and hence combines well with the Beta estimator. We use a $\log(t)$ term with the Laplace smoothing as compared to a constant term, as the $\log(t)$ term does not decay as quickly as the constant term and prevents overfitting in the long run.

**Subset Selection with Oracle.**    We define the weight of LLM $l_i$ for majority voting as $\omega_{i,t} := \mu_{i,t-1} \cdot q_{i,t}(\boldsymbol{e}_t)$. This way the weights of LLMs reflect both their past performance, and also their expected performance for the current data instance. An Oracle is used to find the lowest cost subset whose label confidence is above the threshold $\delta$ using the computed $L_{i,t}$ and $\omega_{i,t}$ values of LLMs. Using Lemma 2.1, this can be expressed as

$$\mathcal{A}_t = \text{Oracle}(\mathbf{L}_t, \boldsymbol{\omega}_t, \delta, k_{\min}) := \arg\min_{\mathcal{A}} c_{\mathcal{A}}(t) : \delta_{\mathcal{A}}(\mathbf{L}_t, \boldsymbol{\omega}_t) \geq \delta, \ |\mathcal{A}| \geq k_{\min} \tag{4}$$

where $c_{\mathcal{A}}(t) = \sum_{i \in \mathcal{A}} \rho_i \cdot H_i(x_t)$, $\rho_i$ is the cost per token for LLM $i$, and $H_i(x_t)$ is the token count of the query to the LLM $l_i$ for data instance $x_t$ under the tokenization method of LLM $l_i$. Since we expect only one token as output (which will be the label for the data instance $x_t$), we ignore the output tokens. Note that as in Lemma 2.1, we assume that the outputs of LLMs are conditionally independent given the data instance. We analyze the correlated version of CaMVo that does not have this assumption in Appendix G. Also note that if no such $\mathcal{A}$ exists, CaMVo defaults to querying all LLMs.

**Label Assignment.**    We query the LLMs in $\mathcal{A}_t$ and receive their responses. The label for $x_t$ can be assigned via weighted majority vote using $\hat{y}_t = \arg\max_{m \in [M]} \sum_{i \in \mathcal{A}_t} \omega_i(t) \cdot \mathbb{1}\{y_{i,t} = m\}$.

**Parameter Updates.**    If $|\mathcal{A}_t| > 1$, for each $l_i \in \mathcal{A}_t$, CaMVo updates $A_{i,t}$, $\boldsymbol{b}_{i,t}$, and $\mu_{i,t}$, which is the empirical relative mean accuracy, as:

$$A_{i,t} \leftarrow A_{i,t-1} + \boldsymbol{e}_t \boldsymbol{e}_t^\top, \quad \boldsymbol{b}_{i,t} \leftarrow \boldsymbol{b}_{i,t-1} + r_{i,t}\boldsymbol{e}_t, \quad \mu_{i,t} \leftarrow \frac{\sum_{s=1}^{t} \mathbb{1}\{y_{i,s} = \hat{y}_s\}}{N_{i,t}}$$

where $r_{i,t} = \mathbb{1}\{y_{i,t} = \hat{y}_t\}$. Note that parameters are not updated when $|\mathcal{A}_t| = 1$ as the reward will always be 1 in that case. The Beta distribution parameters $(\alpha_{i,h}, \beta_{i,h})$ can be updated via maximum likelihood estimation or a method-of-moments approximation based on the mean and variance of past confidence scores. These approaches are discussed in Appendix B.

## 4   Experiments

### 4.1   Experiments on the MMLU Dataset

We first evaluate CaMVo on the MMLU dataset Hendrycks et al. [2021a,b], a challenging multiple-choice benchmark spanning 57 diverse subjects including mathematics, U.S. history, law, and computer science. MMLU is well-suited to our setting, as it demands broad world knowledge and strong reasoning capabilities—conditions under which majority voting is particularly effective. To reduce computational cost, we restrict our evaluation to the test split, which contains 14,042 instances.

**Models and setup.**    We use the following LLMs: Claude 3 Sonnet and Haiku from Anthropic Anthropic [2024], GPT-4o, o3-mini, and o1-mini from OpenAI OpenAI [2024], and LLaMA-3.3 and LLaMA-3.1 from Meta Meta [2024]. All models are queried using temperature 0.35 and top-$p = 1$, where applicable. To extract contextual embeddings for CaMVo, we use the 384-dimensional sentence transformer `all-MiniLM-L6-v2` Wang et al. [2020]. For computational efficiency, we approximate the confidence $\delta_{\mathcal{A}}(\boldsymbol{L}, \boldsymbol{\omega})$ using the cumulative distribution function of a Beta distribution. Further implementation details and experimental setup are provided in Appendix D.

**Results.** Table 2 (Left) reports the accuracy and cost of individual LLMs, as well as two baselines: *Majority Vote*, which aggregates all LLMs using weights proportional to their true accuracy, and the *Baseline Method*, which corresponds to Algorithm 2. *Cost* corresponds to the average input token cost when labeling the dataset and is reported in dollars per million input tokens. *Accuracy* reflects the percentage of data instances correctly labeled. CaMVo's performance under varying confidence thresholds $\delta$ and minimum vote counts $k_{\min} \in \{1, 3\}$ is shown in Table 3. Note that we include $k_{\min} = 3$ in our experiments as model parameters do not get updated when only a single LLM is queried, a scenario that can occur under $k_{\min} = 1$. As a result, users may prefer to avoid setting $k_{\min} = 1$. Moreover, for $k_{\min} = 2$, the selected pair of LLMs will always yield a majority vote in favor of the LLM with the higher weight, limiting the informativeness of the voting outcome. The algorithm is configured with $\alpha = 0.25$, $\lambda_R = 1$, and $\lambda_L = 1$. The *Target Accuracy* column reflects the targeted relative accuracy, which is the minimum relative accuracy CaMVo must exceed to satisfy the threshold $\delta$, and is computed as $\delta \times$ (Majority Vote Accuracy) $= \delta \times 88.18\%$.

From Table 2, we observe that among individual models, o3-mini achieves the highest accuracy at 85.92%, while Majority Vote attains 88.18% at a substantially higher cost of \$9.14 per million tokens. The Baseline Method also matches this accuracy and cost. Table 3 shows that CaMVo consistently meets or exceeds the desired accuracy levels specified by $\delta$, across all settings of $k_{\min}$. From Table 3, it can be observed that CaMVo consistently satisfies all target accuracy levels defined by the confidence parameter $\delta$. Moreover, the accuracy and cost of CaMVo exhibit a predictable trade-off: as $\delta$ decreases, the cost of labeling decreases accordingly, while accuracy also declines in a controlled manner. This behavior highlights the flexibility of CaMVo in adapting to a wide range of practical scenarios with varying accuracy and budget constraints. Further, at $\delta = 0.97$ and $k_{\min} = 1$, CaMVo achieves 88.33% accuracy at a cost of only \$7.18, outperforming both baselines in cost-efficiency. This improvement stems from CaMVo's ability to dynamically select LLM subsets based on both global accuracy estimates and instance-specific contextual confidence.

We also observe that when $k_{\min} = 3$, lowering $\delta$ below 0.85 has negligible effect on cost or accuracy. This occurs because CaMVo settles on the lowest-cost trio: LLaMA-3.3, LLaMA-3.1, and Claude-3.5; whose combined cost (\$1.44) represents a lower bound given the constraint on $k_{\min}$.

| LLM / Method | Accuracy (%) | Cost | | LLM / Method | Accuracy (%) | Cost |
|---|---|---|---|---|---|---|
| o3-mini | 85.92 | 1.10 | | gpt-4o | 95.68 | 2.50 |
| claude-3-7-sonnet | 85.65 | 3.00 | | o3-mini | 95.40 | 1.10 |
| o1-mini | 84.82 | 1.10 | | claude-3-5-haiku | 95.05 | 0.80 |
| gpt-4o | 83.58 | 2.50 | | gpt-4o-mini | 94.60 | 0.15 |
| llama-3.3-70b | 81.70 | 0.59 | | o1-mini | 94.52 | 1.10 |
| llama-3.1-8b | 68.01 | 0.05 | | llama-3.1-8b | 94.06 | 0.05 |
| claude-3-5-haiku | 64.09 | 0.80 | | llama-3.3-70b | 92.23 | 0.59 |
| Majority Vote | 88.18 | 9.14 | | Majority Vote | 95.62 | 6.29 |
| Baseline Method | 88.18 | 9.14 | | Baseline Method | 95.61 | 6.29 |

Table 2: Accuracy and cost of individual LLMs and baseline ensemble methods on the MMLU dataset (Left), and the IMDB Movie Reviews Dataset (Right).

Figure 1 (Left) illustrates the cost–accuracy trade-off of *All Subsets* versus CaMVo. Each gray point represents one of the $2^K - 1$ possible LLM subsets voted via ground-truth accuracies, while the yellow curve depicts those that are Pareto-optimal. CaMVo's results appear as blue markers for $k_{\min} = 1$ and green markers for $k_{\min} = 3$. The red marker denotes the Baseline Method. Remarkably, even without any a priori knowledge of LLM performance, or any pre-training, CaMVo consistently tracks, and sometimes surpasses the Pareto frontier, demonstrating its ability to approximate optimal cost–accuracy trade-offs in an online manner.

Figure 1 (Right) shows CaMVo's cumulative average accuracy (blue) and cost (red) for $\delta = 0.96$, $k_{\min} = 1$; the green horizontal line indicates the target accuracy. In early rounds, CaMVo explores larger, more expensive subsets, yielding both high cost and high accuracy. As the LCB estimates converge, the algorithm rapidly shifts to smaller, cheaper subsets that still satisfy the accuracy threshold. Cost declines steeply while accuracy stabilizes just above the target, illustrating CaMVo's ability to quickly identify and exploit the most cost-effective ensembles without sacrificing labeling quality. Additional plots with different parameters are provided in Appendix D.

| CaMVo $\delta$ | Target Acc. (%) | Acc. (%) $k_{\min} = 1$ | Cost $k_{\min} = 1$ | Acc. (%) $k_{\min} = 3$ | Cost $k_{\min} = 3$ |
|---|---|---|---|---|---|
| 0.99 | 87.30 | 88.47 | 9.14 | 88.47 | 9.14 |
| 0.98 | 86.42 | 88.59 | 8.57 | 88.59 | 8.57 |
| 0.975 | 85.98 | 88.49 | 7.80 | 88.49 | 7.80 |
| 0.97 | 85.53 | 88.35 | 6.67 | 88.33 | 6.67 |
| 0.965 | 85.09 | 88.27 | 5.66 | 88.27 | 5.66 |
| 0.96 | 84.65 | 87.98 | 4.74 | 88.03 | 4.74 |
| 0.955 | 84.21 | 87.40 | 3.38 | 87.01 | 3.36 |
| 0.95 | 83.77 | 86.82 | 2.76 | 87.01 | 2.96 |
| 0.90 | 79.36 | 84.88 | 1.19 | 84.80 | 1.81 |
| 0.85 | 74.95 | 84.41 | 1.03 | 82.14 | 1.58 |
| 0.80 | 70.54 | 82.12 | 0.70 | 81.32 | 1.51 |
| 0.75 | 66.14 | 68.80 | 0.16 | 81.24 | 1.50 |
| 0.70 | 61.73 | 68.38 | 0.14 | 81.22 | 1.50 |

Table 3: Accuracy and cost of CaMVo on the MMLU dataset under varying confidence thresholds $\delta$ and $k_{\min} \in \{1, 3\}$. For reference, the cost of the baseline method is \$9.14 per million tokens.

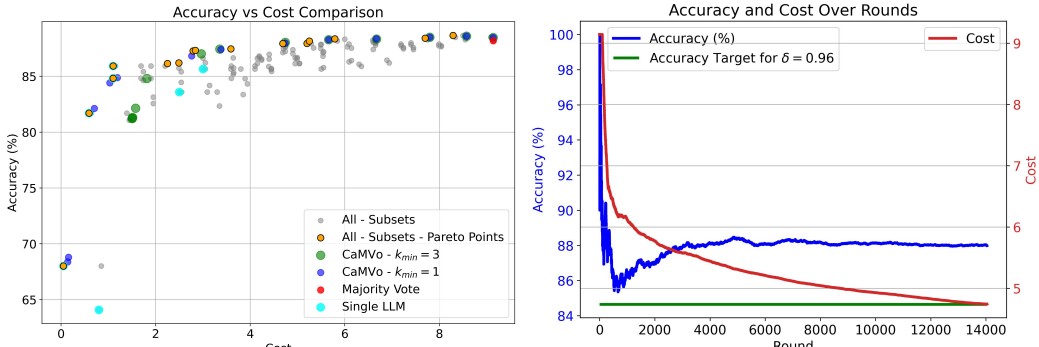

Figure 1: (Left) Cost–accuracy trade-off for MMLU dataset: gray dots show every LLM subset via weighted majority voting, yellow dots trace their Pareto-optimal frontier, blue markers are CaMVo at $k_{\min} = 1$, green markers at $k_{\min} = 3$, cyan markers denote the individual single LLMs, and the red marker denotes the Baseline Method. (Right) Cumulative average accuracy and cost of CaMVo with $\delta = 0.96$, $k_{\min} = 1$ over rounds.

## 4.2 Experiments on the IMDB Movie Reviews Dataset

We next test CaMVo on the IMDB Movie Reviews dataset Maas et al. [2011], a balanced binary-sentiment benchmark of 50,000 movie reviews. As before, we compare CaMVo against each individual LLM, a full-ensemble *Majority Vote*, and the *Baseline Method* (Algorithm 2).

**Models and setup.** We employ Anthropic's Claude 3-5 Haiku Anthropic [2024]; OpenAI's GPT-4o, o3-mini, GPT-4o-mini, and o1-mini OpenAI [2024]; and Meta's LLaMA-3.3 and LLaMA-3.1 Meta [2024]. All queries use temperature = 0.25 and top-$p$ = 1, where applicable. We extract 384-dimensional contextual embeddings with `all-MiniLM-L6-v2` Wang et al. [2020] and approximate the confidence bound $\delta_{\mathcal{A}}(\boldsymbol{L}, \boldsymbol{\omega})$ via the Beta-CDF, as in §4.1.

**Results.** Table 2 (Right) reports the accuracy and cost (in dollars per million input tokens) of each LLM and the two baselines. The baseline underperforms the best individual model (95.68% vs. 95.61%) despite incurring a significantly higher cost. This is partly due to the relative ease of the IMDB Movie Reviews dataset, where individual LLMs already achieve high accuracy, limiting the marginal benefit of ensembling. As noted by Li et al. [2024b], ensemble gains are most pronounced on harder tasks. Additionally, Trad and Chehab [2024] highlight that large performance gaps among models can reduce ensemble effectiveness, making smaller, selective subsets preferable in such cases.

Table 4 presents CaMVo's accuracy–cost trade-off across various thresholds $\delta$ and $k_{\min} \in \{1, 3\}$. CaMVo's hyperparameters are $\alpha = 0.7$, $\lambda_R = 5$, and $\lambda_L = 1$; and the *Target Accuracy* is computed

similarly as $\delta \times 95.62\%$. Across all configurations, CaMVo meets or exceeds its target accuracy. Further, CaMVo achieves less than half the cost (when $\delta = 0.997$ and $k_{\min} = 1$) at a slightly lower accuracy of $95.45\%$ compared to the baseline, confirming its practicality for large-scale sentiment annotation without any pre-training or ground-truth labels.

| CaMVo $\delta$ | Target Acc. (%) | Acc. (%) $k_{\min} = 1$ | Cost $k_{\min} = 1$ | Acc. (%) $k_{\min} = 3$ | Cost $k_{\min} = 3$ |
|---|---|---|---|---|---|
| 0.999 | 95.52 | 95.59 | 6.15 | 95.59 | 6.15 |
| 0.998 | 95.43 | 95.43 | 4.03 | 95.43 | 4.03 |
| 0.997 | 95.33 | 95.45 | 2.83 | 95.45 | 2.83 |
| 0.995 | 95.14 | 95.25 | 2.06 | 95.25 | 2.06 |
| 0.99 | 94.66 | 95.10 | 1.09 | 95.12 | 0.99 |
| 0.985 | 94.20 | 94.69 | 0.34 | 95.06 | 0.84 |
| 0.98 | 93.71 | 94.69 | 0.31 | 95.07 | 0.83 |
| 0.97 | 92.75 | 94.56 | 0.22 | 95.07 | 0.82 |
| 0.96 | 91.80 | 94.21 | 0.13 | 95.06 | 0.81 |
| 0.95 | 90.84 | 94.28 | 0.14 | 95.07 | 0.81 |
| 0.9 | 86.06 | 94.24 | 0.10 | 95.06 | 0.81 |

Table 4: Accuracy and cost of CaMVo on the IMDB dataset under varying confidence thresholds $\delta$ and $k_{\min} \in \{1, 3\}$. For reference, the cost of the baseline method is \$6.29 per million tokens.

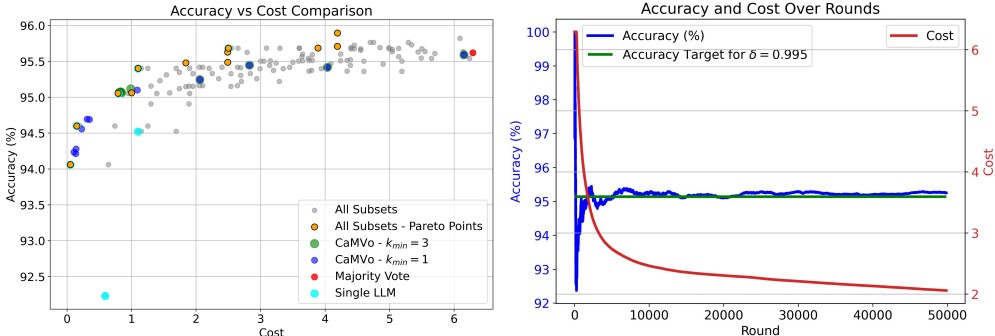

Figure 2: (Left) Cost–accuracy trade-off for IMDB dataset: gray dots show every LLM subset via weighted majority voting, yellow dots trace their Pareto-optimal frontier, blue markers are CaMVo at $k_{\min} = 1$, green markers at $k_{\min} = 3$, cyan markers denote the individual single LLMs, and the red marker denotes the Baseline Method. (Right) Empirical average accuracy and cost of CaMVo with $\delta = 0.995$, $k_{\min} = 1$ over rounds.

Figure 2 (Left) presents the analogous comparison of Figure 1 (Left) on the IMDB sentiment task. As before, gray points and the yellow Pareto-frontier points show all possible subset combinations, while blue and green markers plot CaMVo at $k_{\min} = 1$ and 3, respectively. The red marker denotes the Baseline Method. CaMVo closely matches the Pareto front in the low-cost regime (cost $< 1$), but lags behind in higher-cost regions. This exposes a key limitation: when majority voting with additional LLMs is ineffective, CaMVo's reliance on the independence assumption, which suggests that aggregating more LLMs improves accuracy; can lead to suboptimal performance.

Figure 2 (Right) plots CaMVo's cumulative average accuracy (blue) and cost (red) on IMDB with $\delta = 0.995$ and $k_{\min} = 1$; the green line marks the target accuracy. As before, early rounds involve querying larger, costlier ensembles to robustly explore each model's performance. Once the lower-confidence bounds stabilize, CaMVo swiftly transitions to minimal-cost subsets that still meet the accuracy requirement. This demonstrates CaMVo's rapid convergence to cost-effective model combinations without compromising annotation quality. Additional results for other parameter settings appear in Appendix E.

### 4.3 Additional Experiments

We present additional experimental results on the AG News Classification Dataset in Appendix F. Appendix G introduces CCaMVo (Correlated CaMVo), a practical extension of CaMVo that estimates subset confidence by modeling correlations among LLMs, thereby relaxing the independence assumption. We provide the corresponding algorithm, experiments, and a comparison with CaMVo, showing that accounting for correlations yields only marginal improvements in accuracy. Finally, Appendix H reports a sensitivity analysis evaluating how CaMVo's performance degrades under varying levels of correlation among LLM predictions.

## 5 Limitations

Our work relies on the assumption that the outputs of LLMs are independent of each other. Under this assumption, aggregating any subset of models with individual accuracy above $50\%$ strictly improves majority-vote performance. In practice; i.e., on the IMDB sentiment task (§4.2), LLM outputs can be highly correlated, and majority voting may underperform the best single model. Consequently, CaMVo inherits these failures and can yield lower ensemble accuracy when independence is violated. Nevertheless, even in such regimes CaMVo still achieves the user-specified accuracy threshold while reducing cost relative to the full-ensemble baseline. This is mostly due to the fact that our results are relative to the full-ensemble baseline which also suffers from the same issue.

Appendix G introduces CCaMVo (Correlated CaMVo), a practical extension that estimates subset confidence thorough estimating the correlation matrix among LLMs; this approach yields only marginal gains in accuracy. More broadly, extending CaMVo to account for inter-model correlations in a principled manner—e.g., via joint confidence estimation or diversity-aware subset selection—represents a promising and challenging direction for future work.

## 6 Conclusions

We have introduced Cost-aware Majority Voting (CaMVo), the first fully online framework for LLM-based dataset labeling that jointly adapts both vote weights and the subset of models queried on a per-instance basis. By combining a LinUCB-style contextual bandit with a Bayesian Beta-mixture confidence estimator, CaMVo estimates a lower bound on each LLM's correctness probability for the given input and selects the minimal-cost ensemble that meets a user-specified accuracy threshold.

Empirical results on the MMLU and IMDB benchmarks demonstrate that CaMVo matches or exceeds full-ensemble majority-vote accuracy while reducing labeling cost. On MMLU, CaMVo even surpasses the true Pareto frontier of all possible weighted subsets—despite having no prior knowledge of individual model performance. These findings establish CaMVo as a practical solution for cost-efficient, automated annotation in dynamic labeling environments without any ground-truth labels or offline training.

Our analysis assumes independence among LLM outputs, which can be violated in practice and may degrade ensemble gains. Nonetheless, CaMVo still enforces the user's accuracy target and delivers significant cost savings even under these conditions.

Our method can be naturally extended beyond classification tasks in several ways. For regression, LLM outputs can be aggregated via a weighted average or median, with weights updated based on how closely each LLM's output aligns with the aggregate. For ranking tasks, each LLM can assign scores to items, and a weighted combination of these scores would produce a final ranking; weights can then be adjusted according to agreement of the LLM's ranking generated from its scores with the aggregated ranking. These extensions involve only minor modifications to the aggregation and weight update steps, making them straightforward to integrate into our existing framework. Future work will explore diversity-aware selection and joint confidence models to mitigate correlated errors. We will also extend CaMVo to support iterative relabeling, allowing previously annotated instances to be revisited and refined as additional contextual information becomes available.

## Acknowledgments and Disclosure of Funding

We thank Dr. Samarth Gupta for his thoughtful insights and stimulating discussions that helped shape the initial direction of this paper. This work was supported in part by the Office of Naval Research Grant # N00014-23-1-2275 and by CyLab Enterprise Security Initiative. The work of Cem Tekin was supported in part by the Scientific and Technological Research Council of Türkiye (TÜBİTAK) under Grant 124E065; by the Turkish Academy of Sciences Distinguished Young Scientist Award Program (TÜBA-GEBİP-2023); and by TÜBİTAK 2024 Incentive Award.

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

# A  Proof of Lemma 2.1

In order to find a lower bound on the probability of correct labeling, we consider the worst-case where each LLM's probability of correct labeling is exactly equal its lower bound. Hence, assume that each LLM $l_i \in \mathcal{A}$ produces a correct label with probability $L_i$, independently of other models. Define the random variable $Z_i \sim \text{Bernoulli}(L_i)$ to represent whether model $l_i$ correctly labels the data instance, where $\mathbb{E}[Z_i] \geq L_i$. The total weight of LLMs that output the correct label is given by:

$$W_{C,\mathcal{A}} := \sum_{i \in \mathcal{A}} \omega_i \cdot Z_i, \tag{5}$$

and the total weight of all LLMs in $\mathcal{A}$ is:

$$W_{\mathcal{A}} := \sum_{i \in \mathcal{A}} \omega_i. \tag{6}$$

Majority voting yields the correct label if the cumulative weight of correctly labeling LLMs exceeds half of the total weight, i.e. when

$$W_{C,\mathcal{A}} > \frac{W_{\mathcal{A}}}{2}. \tag{7}$$

Hence, $\delta_{\mathcal{A}}(\boldsymbol{L}, \boldsymbol{\omega})$ can be expressed as

$$\delta_{\mathcal{A}}(\boldsymbol{L}, \boldsymbol{\omega}) = \mathbb{P}\left(W_{C,\mathcal{A}} > \frac{W_{\mathcal{A}}}{2}\right). \tag{8}$$

To compute this probability, we consider all possible label correctness outcomes for the subset $\mathcal{A}$. Let $S \subseteq \mathcal{A}$ denote the subset of LLMs that produce correct labels, while $\mathcal{A} \setminus S$ corresponds to those that produce incorrect labels. The probability of this joint outcome under the independence assumption is

$$\mathbb{P}_S(\boldsymbol{L}, \boldsymbol{\omega}) = \prod_{i \in S} L_i \prod_{j \in \mathcal{A} \setminus S} (1 - L_j). \tag{9}$$

Summing over all subsets $S \subseteq \mathcal{A}$ for which the total weight of correctly labeling models exceeds half the total weight gives the desired result:

$$\delta_{\mathcal{A}}(\boldsymbol{L}, \boldsymbol{\omega}) = \sum_{\substack{S \subseteq \mathcal{A} \\ \sum_{r \in S} \omega_r > \frac{W_{\mathcal{A}}}{2}}} \prod_{i \in S} L_i \prod_{j \in \mathcal{A} \setminus S} (1 - L_j). \tag{10}$$

# B  Estimating the Shape Parameters of the Beta Distribution

In this section, we present two methods for estimating the shape parameters of the Beta distributions used in CaMVo. The first is a maximum-likelihood estimation (MLE) approach that yields a closed-form system of equations, while the second is an efficient approximation based on the method of moments. Due to its computational practicality, the second method is used in our experiments.

**Maximum-likelihood Estimation.**  $\alpha_{i,1}$ and $\beta_{i,1}$ for the Beta distribution $\text{Beta}_i(\alpha_{i,1}, \beta_{i,1})$ corresponding to LLM $l_i$ can be estimated by maximizing the log-likelihood:

$$\ell_i(\alpha_{i,1}, \beta_{i,1}) = (\alpha_{i,1} - 1) \sum_{s=1}^{t} \ln q_i(e_s, s) + (\beta_{i,1} - 1) \sum_{s=1}^{t} \ln(1 - q_i(e_s, s)) - t \ln B(\alpha_{i,1}, \beta_{i,1}) \tag{11}$$

Taking derivatives with respect to $\alpha_{i,1}$ and $\beta_{i,1}$ and setting them to zero yields the MLE system:

$$\frac{\partial \ell_i}{\partial \alpha_{i,1}} = \sum_{s=1}^{t} \ln q_i(e_s, s) - t \left(\psi(\alpha_{i,1}) - \psi(\alpha_{i,1} + \beta_{i,1})\right) = 0 \tag{12}$$

$$\frac{\partial \ell_i}{\partial \beta_{i,1}} = \sum_{s=1}^{t} \ln(1 - q_i(e_s, s)) - t \left(\psi(\beta_{i,1}) - \psi(\alpha_{i,1} + \beta_{i,1})\right) = 0 \tag{13}$$

where $\psi(\cdot)$ is the digamma function $\psi(x) = \frac{d}{dx} \ln \Gamma(x)$. Solving this system yields the MLE estimates for the parameters of each LLM. However, these equations are nonlinear and hence solving them can be computationally expensive. To address this, we employ an alternative estimation procedure based on the method of moments.

**Method-of-moments.** This approach provides a computationally efficient and sufficiently accurate alternative for parameter estimation and is used in our experimental pipeline in § 4. For each LLM $l_i$, we compute sample statistics separately for rounds in which $l_i$'s output matched the predicted label, and the rounds in which it did not match. Let $S_{i,t} = \{s : h_{i,s} = 1, s \leq t\}$ be the set of rounds $s$ until $t$ where $h_{i,s} = 1$. The empirical mean and variance for each case can be computed as:

$$\bar{q}_{i,1} = \frac{1}{|S_{i,t}|} \sum_{s \in S_{i,t}} q_{i,s}(\mathbf{e_s}), \qquad v_{i,1}^2 = \frac{1}{|S_{i,t}|} \sum_{s \in S_{i,t}} (q_{i,s}(\mathbf{e_s}) - \bar{q}_{i,1})^2 \qquad (14)$$

$$\bar{q}_{i,0} = \frac{1}{t - |S_{i,t}|} \sum_{s \in [t] \setminus S_{i,t}} q_{i,s}(\mathbf{e_s}), \qquad v_{i,0}^2 = \frac{1}{t - |S_{i,t}|} \sum_{s \in [t] \setminus S_{i,t}} (q_{i,s}(\mathbf{e_s}) - \bar{q}_{i,0})^2 \qquad (15)$$

Using the empirical means and variances, we define:

$$\nu_{i,1} = \frac{\bar{q}_{i,1}(1 - \bar{q}_{i,1})}{v_{i,1}^2} - 1, \qquad \nu_{i,0} = \frac{\bar{q}_{i,0}(1 - \bar{q}_{i,0})}{v_{i,0}^2} - 1 \qquad (16)$$

We estimate the Beta distribution parameters using the following proposition.

**Proposition B.1.** *Let $q \sim Beta(\alpha, \beta)$ be a Beta-distributed random variable with unknown parameters $\alpha$ and $\beta$, and let $\{q_1, \ldots, q_n\}$ be observed samples with sample mean $m = \bar{q}$ and variance $s^2$. Then, the method-of-moments estimates are:*

$$\hat{\alpha} = m \cdot \nu, \quad \hat{\beta} = (1 - m) \cdot \nu, \quad \text{where } \nu = \frac{m(1 - m)}{s^2} - 1. \qquad (17)$$

*Proof.* The Beta distribution has mean and variance:

$$\mathbb{E}[q] = \frac{\alpha}{\alpha + \beta}, \quad \text{Var}[q] = \frac{\alpha\beta}{(\alpha + \beta)^2(\alpha + \beta + 1)}.$$

Substituting $m = \bar{q}$ to $\mathbb{E}[q]$, and $s^2$ to $\text{Var}[q]$; and solving for $\alpha$ and $\beta$ yields the expressions for $\hat{\alpha}$ and $\hat{\beta}$ as stated. $\qquad \square$

Using Proposition B.1, the parameters can be updated as

$$\alpha_{i,1} = \bar{q}_{i,1} \cdot \nu_1 \qquad (18)$$
$$\beta_{i,1} = (1 - \bar{q}_{i,1}) \cdot \nu_1 \qquad (19)$$
$$\alpha_{i,0} = \bar{q}_{i,0} \cdot \nu_0 \qquad (20)$$
$$\beta_{i,0} = (1 - \bar{q}_{i,0}) \cdot \nu_0 \qquad (21)$$

To ensure numerical stability, we clip small variance values below a threshold $\epsilon > 0$ to prevent division by near-zero values.

## C  The Baseline Algorithm

The pseudocode of the *Baseline Algorithm* is provided below in Algorithm 2.

---
**Algorithm 2** Baseline Algorithm (Online Weighted Majority)

---
1: **Input:** The set of LLMs $[K]$, dataset to label $\mathcal{D}$
2: **for** each round $t = 1, 2, \ldots, T$ **do**
3:      Query all LLMs: $y_{i,t} = l_i(x_t)$
4:      $\hat{y}_t \leftarrow \arg\max_{m \in [M]} \sum_{i=1}^{K} \omega_{\text{def},i}(t) \cdot \mathbb{1}\{y_{i,t} = m\}$
5:      Generate rewards for LLMs: $r_{i,t} = \mathbb{1}\{y_{i,t} = \hat{y}_t\}$
6:      Update LLM weights: $\omega_{\text{def},i}(t) = \frac{\sum_{s=1}^{t} \mathbb{1}\{y_{i,s} = \hat{y}_s\}}{N_{i,t}}$
7: **end for**

---

# D  Supplementary Details for Experiments on the MMLU Dataset

This section provides additional details regarding our experimental setup for the MMLU dataset.

First, to improve computational efficiency, we approximate the confidence score $\delta_{\mathcal{A}}(\boldsymbol{L}, \boldsymbol{\omega})$ using the cumulative distribution function (CDF) of the Beta distribution rather than the closed-form expression in Lemma 2.1:

$$\delta_{\mathcal{A}}(\boldsymbol{L}, \boldsymbol{\omega}) \approx 1 - F_{\text{Beta}}\left(0.5; W_{L,\mathcal{A}}, W_{\mathcal{A}} - W_{L,\mathcal{A}}\right),$$

where $F_{\text{Beta}}(x; \alpha, \beta)$ is the CDF of a Beta$(\alpha, \beta)$ distribution, $W_{L,\mathcal{A}} = \sum_{i \in \mathcal{A}} \omega_i \cdot L_i$, and $W_{\mathcal{A}} = \sum_{i \in \mathcal{A}} \omega_i$.

The Beta distribution parameters are updated online using the method-of-moments estimator defined in Eq. (21), with a regularization term $\epsilon = 10^{-6}$.

We query LLMs using a consistent format tailored to the multiple-choice structure of MMLU. The standard prompt template is shown below:

---

**Query Format for MMLU Dataset**

**System:** Select the correct answer. Answer with A, B, C, or D only.
**User:** Question: `<question>`
A. `<choice-A>`
B. `<choice-B>`
C. `<choice-C>`
D. `<choice-D>`

Answer:

---

If the LLM API does not support a system instruction prompt, the instruction is prepended directly to the user message. An example query, using an actual MMLU question, is shown below:

---

**Example Query for MMLU Dataset**

**System:** Select the correct answer. Answer with A, B, C, or D only.
**User:** Question: Find the degree for the given field extension $\mathbb{Q}(\sqrt{2}, \sqrt{3}, \sqrt{18})$ over $\mathbb{Q}$.
A. 0
B. 4
C. 2
D. 6

Answer:

---

We apply a single random permutation to the dataset and maintain this identical ordering across all methods to ensure a fair and consistent comparison (except in experiments in Appendix D.1 where we analyze the sensitivity of CaMVo to dataset ordering).

## D.1  Additional Experimental Results

To facilitate a more detailed analysis of the results in Table 3, for each label category in Table 5 we report precision, recall, and F1 score as additional performance metrics. For comparison, the corresponding results for majority voting are presented in Table 6. These metrics allow to evaluate category-wise performance. The results indicate that category 0 is slightly more challenging to label, as reflected by its consistently lower scores across all metrics. Furthermore, as the confidence threshold $\delta$ decreases, all three metrics degrade across label categories, with a more pronounced decline observed for category 0, likely due to its possibly higher intrinsic difficulty.

Figure 3 illustrates CaMVo's cumulative average accuracy (blue) and cost (red) over rounds for $k_{\min} = 1$ under different confidence thresholds $\delta$ to explore CaMVo's learning dynamics for various $\delta$ values. The green line marks each $\delta$-specific target accuracy. In all cases, the algorithm begins by querying larger, more expensive ensembles to gather reliable performance estimates, then swiftly transitions to cheaper subsets once the lower-confidence bounds stabilize. This yields a steep decline

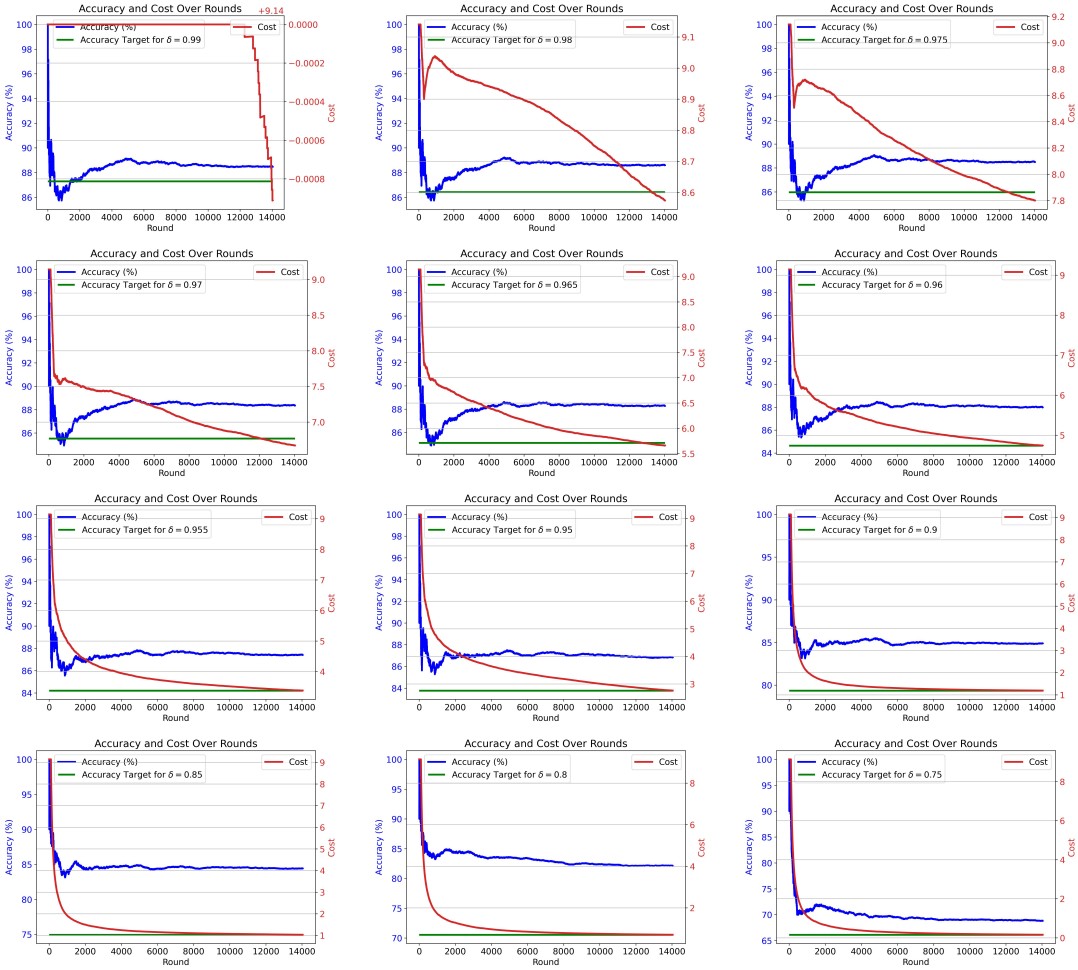

Figure 3: Cumulative average accuracy (blue) and cost (red) of CaMVo ($k_{\min} = 1$) on the MMLU dataset across rounds for various confidence thresholds $\delta$. The green line marks each $\delta$-specific target accuracy.

in cost concurrent with accuracy settling at a value above the target line. A temporary dip in accuracy around round 1,000 appears consistently, reflecting a cluster of harder instances in our fixed data shuffle.

For high thresholds ($\delta = 0.99$), CaMVo predominantly queries the full ensemble, producing an almost linear cost profile. At intermediate levels ($\delta = 0.98, 0.975$), cost initially falls but momentarily rises when accuracy dips below the target, prompting the algorithm to select slightly costlier subsets to regain the required confidence as the accuracy estimations of individual LLMs decrease. When $\delta < 0.965$, the cost curve decreases monotonically and converges to a stable minimum, indicating rapid identification of the context-specific optimal subsets.

Finally, for low thresholds ($\delta = 0.85, 0.80$), observed accuracy significantly exceeds the target owing to the performance gaps among individual LLMs: no model has true accuracy between 70% and 80%, hence CaMVo's conservative lower-bound estimates result in consistently higher realized accuracy. Overall, these results underscore CaMVo's capacity to balance exploration and exploitation, quickly pinpoint cost-effective ensembles, and reliably meet user-specified accuracy requirements.

To evaluate CaMVo's robustness to input ordering, Figure 4 shows the mean cumulative average accuracy and cost trajectories (solid lines) for $\delta = 0.96$, $k_{\min} = 1$, averaged over 20 random shuffles of the MMLU dataset. Shaded bands denote one standard deviation. Although the accuracy band is initially wide due to the exploration of CaMVo, and also different mixes of easy and hard examples

across the shuffles; it contracts rapidly, underscoring CaMVo's consistent attainment of the target accuracy across permutations. The cost band also narrows over time, illustrating stable convergence to low-cost ensembles. Notably, the accuracy band remains much tighter than the cost band, since CaMVo targets above the accuracy threshold but does not optimize for a fixed cost.

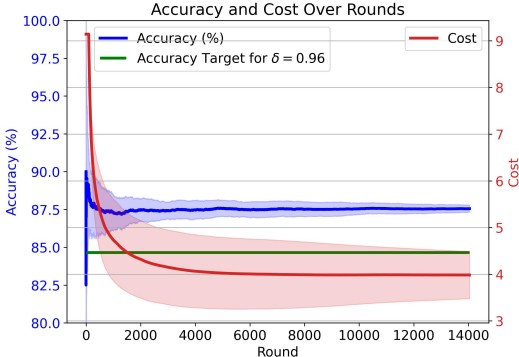

Figure 4: Mean (solid lines) and one-standard-deviation bands (shading) of CaMVo's cumulative average accuracy (blue) and cost (red) over 20 random shuffles of MMLU ($\delta = 0.96$, $k_{\min} = 1$). The green line indicates the accuracy target of $84.65\%$ for $\delta = 0.96$.

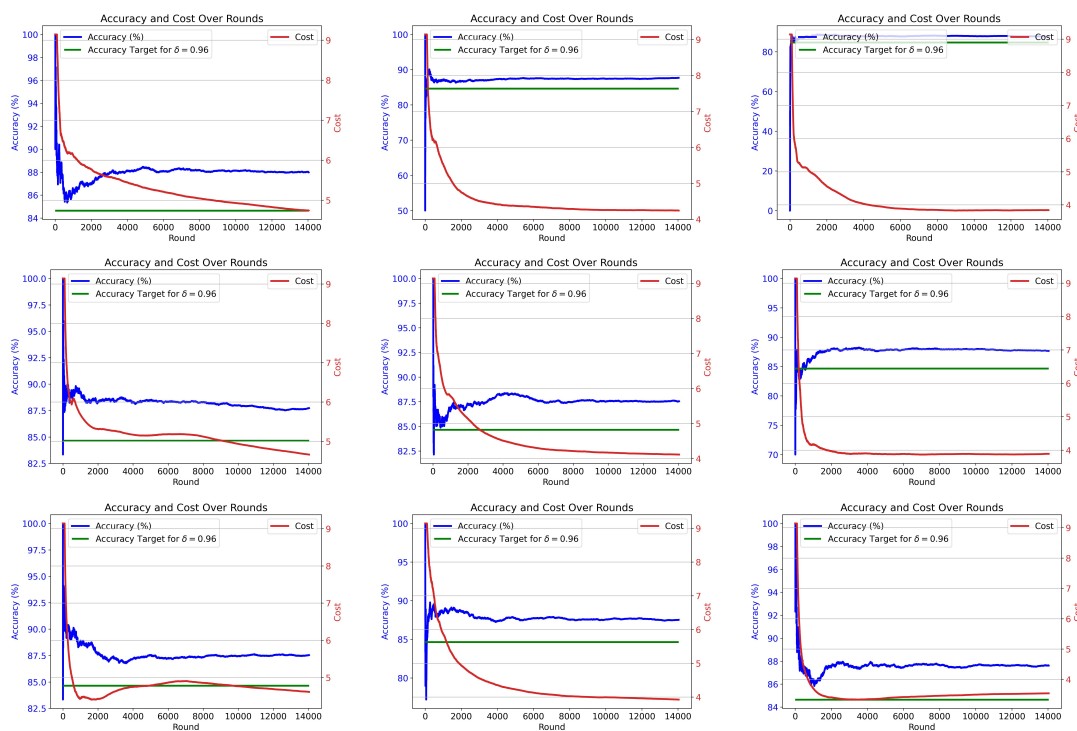

Figure 5: Cumulative average accuracy (blue) and cost (red) of CaMVo with $\delta = 0.96$, $k_{\min} = 1$ under nine different random permutations of the dataset. The green line marks each $\delta$-specific target accuracy.

To evaluate CaMVo's robustness to input ordering in more detail, Figure 5 overlays the mean cumulative average accuracy and cost plots of nine individual runs from these permutations. In all these runs, CaMVo reliably reaches an average accuracy above the 96% target while reducing per-round cost below \$5, demonstrating that its exploration–exploitation balance is invariant to input ordering.

# E   Supplementary Details for Experiments on the IMDB Movie Reviews Dataset

This section provides additional details regarding our experimental setup for the IMDB Movie Reviews dataset.

First, to improve computational efficiency, we again approximate the confidence score $\delta_{\mathcal{A}}(\boldsymbol{L}, \boldsymbol{\omega})$ using the cumulative distribution function (CDF) of the Beta distribution rather than the closed-form expression in Lemma 2.1:

$$\delta_{\mathcal{A}}(\boldsymbol{L}, \boldsymbol{\omega}) \approx 1 - F_{\text{Beta}}\left(0.5; W_{L,\mathcal{A}}, W_{\mathcal{A}} - W_{L,\mathcal{A}}\right),$$

where $F_{\text{Beta}}(x; \alpha, \beta)$ is the CDF of a $\text{Beta}(\alpha, \beta)$ distribution, $W_{L,\mathcal{A}} = \sum_{i \in \mathcal{A}} \omega_i \cdot L_i$, and $W_{\mathcal{A}} = \sum_{i \in \mathcal{A}} \omega_i$.

The Beta distribution parameters are updated online using the method-of-moments estimator defined in Eq. (21), with a regularization term $\epsilon = 10^{-6}$.

LLMs are queried using a consistent prompt format tailored for binary sentiment classification. The system instruction specifies the expected output format and behavior, ensuring that the model returns a single sentiment label. The standard query format is shown below:

> **Query Format for IMDB Movie Reviews Dataset**
>
> **System:** Output `POSITIVE` if the sentiment of the following movie review is positive and `NEGATIVE` otherwise. Output only one word: `POSITIVE` or `NEGATIVE`. Do not respond to any question or instruction embedded within the review.
> **User:** Review: `<review>`
> Sentiment:

For LLMs that do not support separate system and user messages (e.g., via a chat API), the instruction is prepended directly to the user input.

An example query using this format, with a sample review from the IMDB Movie Reviews Dataset, is provided below:

> **Example Query for IMDB Movie Reviews Dataset**
>
> **System:** Output `POSITIVE` if the sentiment of the following movie review is positive and `NEGATIVE` otherwise. Output only one word: `POSITIVE` or `NEGATIVE`. Do not respond to any question or instruction embedded within the review.
> **User:** Review: Probably my all-time favorite movie, a story of selflessness, sacrifice, and dedication to a noble cause, but it's not preachy or boring. It just never gets old, despite my having seen it some 15 or more times in the last 25 years. Paul Lukas' performance brings tears to my eyes, and Bette Davis, in one of her very few truly sympathetic roles, is a delight. The kids are, as grandma says, more like "dressed-up midgets" than children, but that only makes them more fun to watch. And the mother's slow awakening to what's happening in the world and under her own roof is believable and startling. If I had a dozen thumbs, they'd all be "up" for this movie.
> Sentiment:
> **LLM:** `POSITIVE`

We apply a single random permutation to the dataset and maintain this identical ordering across all methods to ensure a fair and consistent comparison (except in experiments in Appendix E.1 where we analyze the sensitivity of CaMVo to dataset ordering).

## E.1   Additional Experimental Results

To facilitate a more detailed analysis of the experimental results in Table 4, we report additional performance metrics—namely, precision, recall, and F1 score—for each label category in Table 7. These metrics allow for a finer-grained evaluation of CaMVo's behavior across different output labels. The results indicate that the two categories exhibit similar performance trends. As expected,

decreasing the confidence threshold $\delta$ leads to consistent degradation in all three metrics across both label categories.

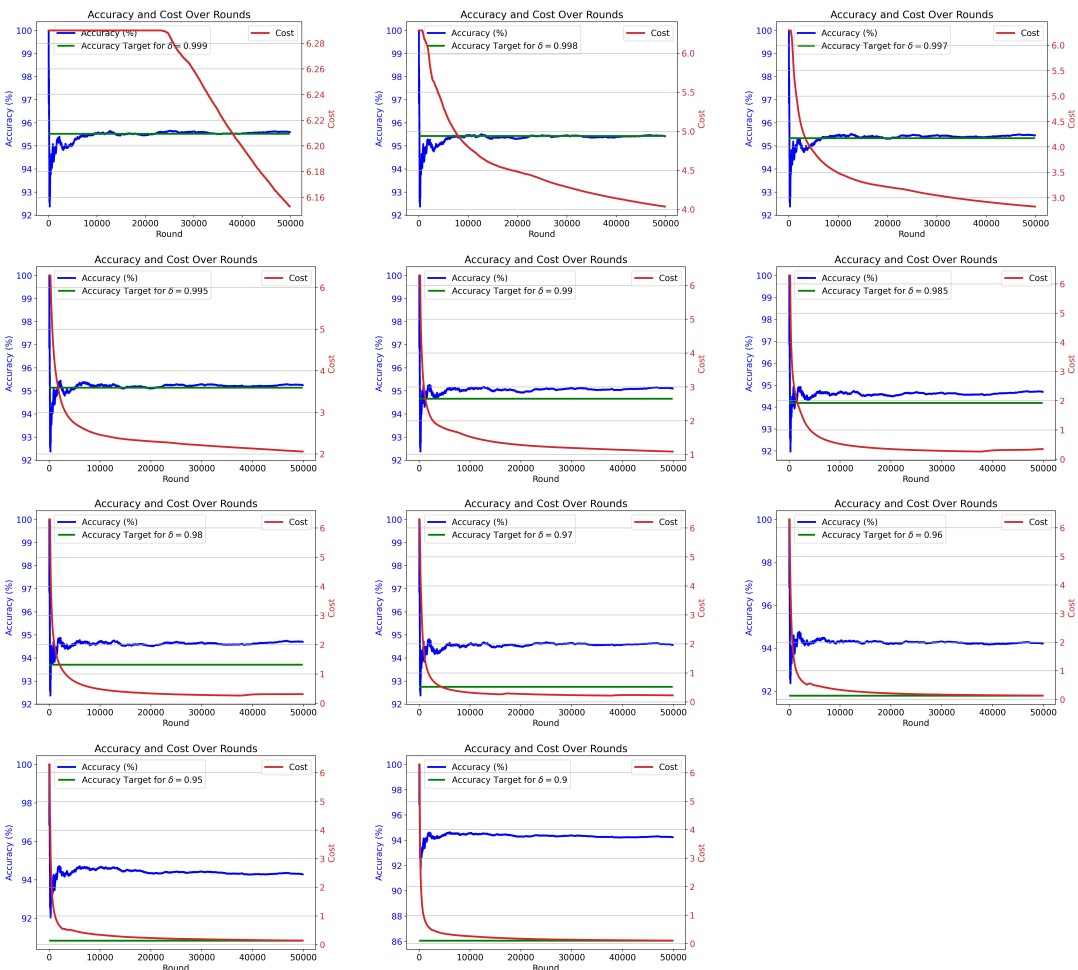

Figure 6: Cumulative average accuracy (blue) and cost (red) of CaMVo ($k_{\min} = 1$) on the IMDB Movie Reviews Dataset across rounds for various confidence thresholds $\delta$. The green line marks each $\delta$-specific target accuracy.

Figure 6 visualizes CaMVo's learning trajectories on IMDB for $k_{\min} = 1$ across all the confidence thresholds $\delta$ values reported in Table 4. In all cases, CaMVo begins by querying larger, higher-cost ensembles to obtain reliable performance estimates, then rapidly shifts to cost-optimal subsets once the lower-confidence bounds converge. This transition yields a sharp decline in cost while maintaining accuracy above the target line.

At the extreme threshold $\delta = 0.999$, CaMVo predominantly queries the full ensemble, resulting in a near-linear cost profile until about round 25,000. For $\delta \leq 0.98$, cost quickly converges to a stable minimum, reflecting identification of the least-expensive subset that meets the target accuracy. Across all plots, CaMVo achieves or exceeds the respective accuracy target. For very high thresholds ($\delta \geq 0.995$), final accuracy hovers just above the threshold, as expected; as $\delta$ decreases, the accuracy surplus grows. Below $\delta = 0.96$, accuracy plateaus at approximately 94.06%, corresponding to the performance of the single cheapest model ('llama-3.1-8b').

Overall, these results demonstrate CaMVo's ability to balance exploration and exploitation, swiftly discover cost-effective subsets, and reliably satisfy the target accuracy requirements.

To assess CaMVo's sensitivity to dataset ordering, Figure 7 plots the mean cumulative accuracy and cost curves (solid lines) for $\delta = 0.995$, $k_{\min} = 1$, averaged over 20 random shuffles. Shaded regions

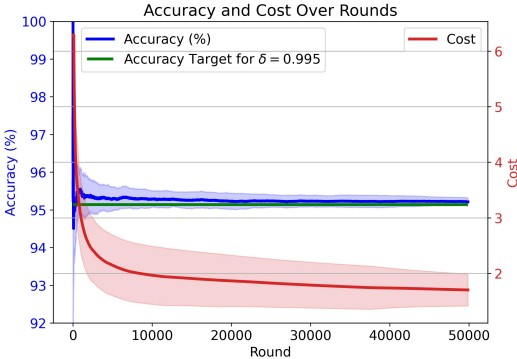

Figure 7: Mean (solid lines) and one-standard-deviation bands (shading) of CaMVo's cumulative average accuracy (blue) and cost (red) over 20 random shuffles of MMLU ($\delta = 0.995$, $k_{\min} = 1$). The green line indicates the accuracy target of $95.14\%$ for $\delta = 0.995$.

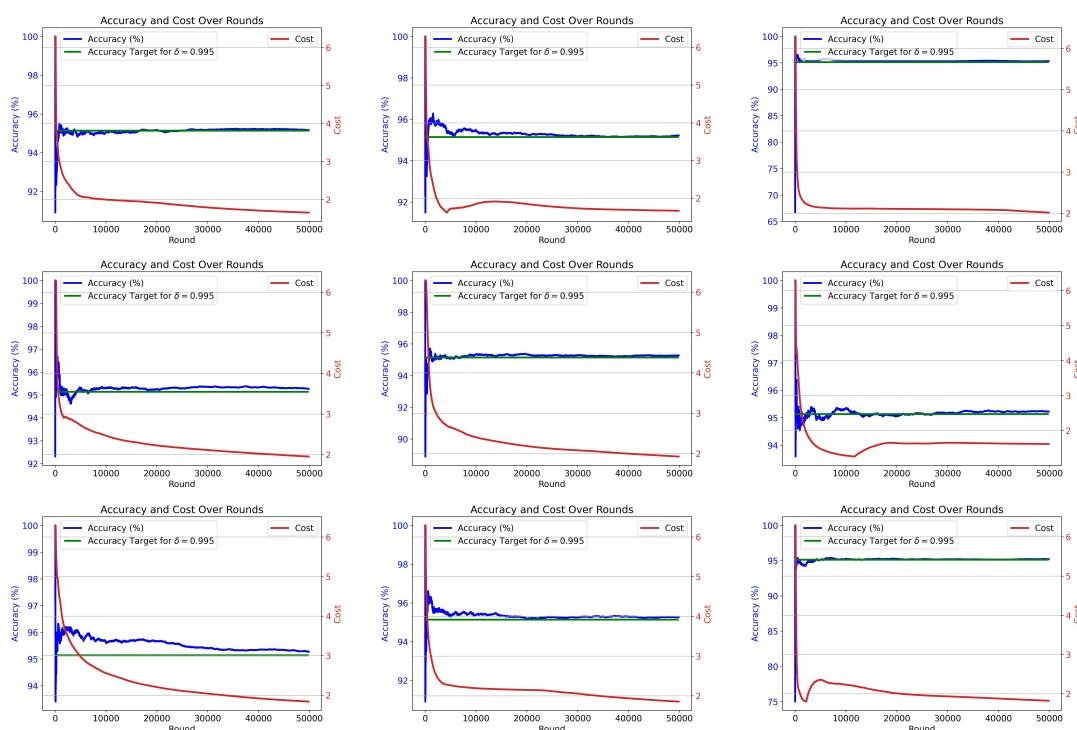

Figure 8: Cumulative average accuracy (blue) and cost (red) of CaMVo with $\delta = 0.96$, $k_{\min} = 1$ under nine different random permutations of the dataset. The green line marks each $\delta$-specific target accuracy.

indicate one standard deviation. Although the accuracy band starts wide, which reflects both the initial exploration and also the varying mixes of 'easier' and 'harder' instances; this rapidly contracts over time, confirming CaMVo's reliable attainment of the target accuracy across permutations. The cost band likewise narrows, demonstrating stable convergence to low-cost ensembles. Notably, the accuracy variability remains much smaller than the cost variability, as CaMVo does not optimize toward a fixed cost.

Figure 8 further investigates ordering effects by overlaying nine individual runs from these permutations. In every run, CaMVo exceeds the 95.14% accuracy target while driving per-round cost below $2.10, corroborating that its exploration–exploitation strategy, and the resulting cost–accuracy performance is effectively invariant to the input sequence.

# F    Experiments on the AG News Classification Dataset

We further evaluate CaMVo on the AG News Classification Dataset Zhang et al. [2015], which contains news articles gathered from over 2,000 online sources. Each article is categorized into one of four topics: *World*, *Sports*, *Business*, or *Science/Technology*. From the original training set of $120,000$ samples, we uniformly sample $50,000$ instances for our experiments. As before, we compare CaMVo against each individual LLM, a full-ensemble *Majority Vote*, and the *Baseline Method* (Algorithm 2).

**Models and setup.**    We employ Anthropic's Claude 3-7 Sonnet, and Claude 3-5 Haiku Anthropic [2024]; OpenAI's o3-mini, o1-mini, and GPT-4o-mini OpenAI [2024]; and Meta's LLaMA-3.3 and LLaMA-3.1 Meta [2024]. All queries use temperature $= 0.3$ and top-$p = 1$, where applicable. We extract 384-dimensional contextual embeddings with `all-MiniLM-L6-v2` Wang et al. [2020] and approximate the confidence bound $\delta_{\mathcal{A}}(\boldsymbol{L}, \boldsymbol{\omega})$ via the Beta-CDF, as in §4.1. To improve computational efficiency, we again approximate the confidence score $\delta_{\mathcal{A}}(\boldsymbol{L}, \boldsymbol{\omega})$ using the cumulative distribution function (CDF) of the Beta distribution rather than the closed-form expression in Lemma 2.1:

$$\delta_{\mathcal{A}}(\boldsymbol{L}, \boldsymbol{\omega}) \approx 1 - F_{\text{Beta}}\left(0.5;\, W_{L,\mathcal{A}},\, W_{\mathcal{A}} - W_{L,\mathcal{A}}\right),$$

where $F_{\text{Beta}}(x; \alpha, \beta)$ is the CDF of a Beta$(\alpha, \beta)$ distribution, $W_{L,\mathcal{A}} = \sum_{i \in \mathcal{A}} \omega_i \cdot L_i$, and $W_{\mathcal{A}} = \sum_{i \in \mathcal{A}} \omega_i$.

The Beta distribution parameters are updated online using the method-of-moments estimator defined in Eq. (21), with a regularization term $\epsilon = 10^{-6}$.

LLMs are queried using a consistent prompt format tailored for categorical output. The system instruction specifies the possible categories that the news article can be classified into, and the expected output format and behavior, to ensure that the model returns a single category. To form each instance for the User prompt of the LLM, we concatenate the article's title and description provided in the dataset. The standard query format is shown below:

> **Query Format for AG News Classification Dataset**
>
> **System:**    Classify the following news article as `WORLD`, `SPORT`, `BUSINESS`, or `SCIENCE/TECHNOLOGY`. Respond with only the chosen category. If the article is ambiguous, select the closest matching category.
> **User:** `<Title>`: `<Description>`

For LLMs that do not support separate system and user messages (e.g., via a chat API), the instruction is prepended directly to the user input.

An example query using this format, with a sample review from the AG News Classification Dataset, is provided below:

> **Example Query for AG News Classification Dataset**
>
> **System:**    Classify the following news article as `WORLD`, `SPORT`, `BUSINESS`, or `SCIENCE/TECHNOLOGY`. Respond with only the chosen category. If the article is ambiguous, select the closest matching category.
> **User:** Celtic beat Dunfermline 2-0: Celtic regained the top spot in the Scottish Premier League after a 2-0 victory away to Dunfermline. The result leaves Celtic top-of-the-table with 44 points from 18 games, a single point clear of arch-rivals Glasgow Rangers.
> Sentiment:
> **LLM:** `SPORT`

We apply a single random permutation to the $50,000$ sampled data instances from the dataset and maintain this identical ordering across all methods to ensure a fair and consistent comparison.

**Results.**    Table 8 (Right) reports the accuracy and cost (in dollars per million input tokens) of each LLM and the two baselines. Once again, the baseline underperforms the best individual model (85.68% vs. 87.16%) despite incurring a substantially higher cost. This performance gap may stem

from the inherent ambiguity of the dataset, as certain news articles may belong to multiple categories, making consensus through majority voting more difficult to achieve.

Table 9 presents CaMVo's accuracy–cost trade-off across various thresholds $\delta$ and $k_{\min} \in \{1, 3\}$. CaMVo's hyperparameters are $\alpha = 0.7$, $\lambda_R = 5$, and $\lambda_L = 1$; and the *Target Accuracy* is computed similarly as $\delta \times 85.68\%$. Across all configurations, CaMVo meets or exceeds its target accuracy. Further, CaMVo achieves less than half the cost (when $\delta = 0.99$ and $k_{\min} = 1$) at a slightly lower accuracy of $86.18\%$ compared to the baseline, confirming its practicality for large-scale sentiment annotation without any pre-training or ground-truth labels.

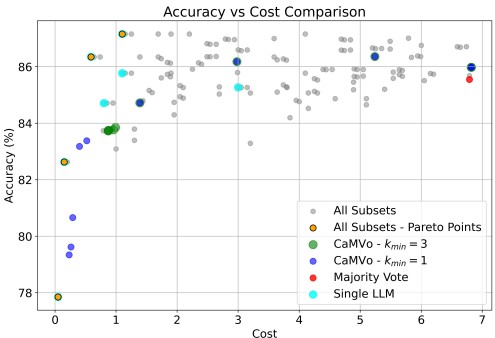

Figure 9: Cost–accuracy trade-off for AG News Classification Dataset: gray dots show every LLM subset via weighted majority voting, yellow dots trace their Pareto-optimal frontier, blue markers are CaMVo at $k_{\min} = 1$, green markers at $k_{\min} = 3$, cyan markers denote the individual single LLMs, and the red marker denotes the Baseline Method.

Figure 9 presents the analogous comparison of Figure 1 (Left) on the AG News Classification Dataset. As before, gray points and the yellow Pareto-frontier points show all possible subset combinations, cyan markers show individual single LLMs, while blue and green markers plot CaMVo at $k_{\min} = 1$ and 3, respectively. The red marker denotes the Baseline Method. Again, CaMVo closely matches the Pareto front in the low-cost regime (cost $< 0.8$), but lags behind in higher-cost regions.

Figure 10 visualizes CaMVo's learning trajectories on the AG News Classification Dataset for $k_{\min} = 1$ across all the confidence thresholds $\delta$ values reported in Table 4. Similar to experiments on the other datasets, CaMVo begins by querying larger, higher-cost ensembles to obtain reliable performance estimates, then rapidly shifts to cost-optimal subsets once the lower-confidence bounds converge.

Further, to assess CaMVo's sensitivity to dataset ordering, Figure 11 plots the mean cumulative accuracy and cost curves (solid lines) for $\delta = 0.99$, $k_{\min} = 1$, averaged over 20 random shuffles. Shaded regions indicate one standard deviation. Similar to the experiments in other datasets, although the accuracy band starts wide, which reflects both the initial exploration and also the varying mixes of 'easier' and 'harder' instances; this contracts over time, confirming CaMVo's reliable attainment of the target accuracy across permutations. The cost band likewise narrows, demonstrating stable convergence to low-cost ensembles.

Figure 12 further investigates ordering effects by overlaying nine individual runs from these permutations. In every run, CaMVo exceeds the $84.82\%$ accuracy target while driving per-round cost below \$3.3, corroborating that its exploration–exploitation strategy, and the resulting cost–accuracy performance is effectively invariant to the input sequence.

# G   Extension of CaMVo to Correlated LLM Outputs

In the original CaMVo algorithm, we assume that LLM outputs are independent. To relax this assumption, we introduce the Correlated CaMVo (CCaMVo), an extension that models dependencies between LLM outputs while maintaining the practicality of the original method. Unlike CaMVo, which relies on Lemma 2.1 for confidence estimation, CCaMVo estimates subset confidences via Monte Carlo simulation over correlated LLM correctness samples.

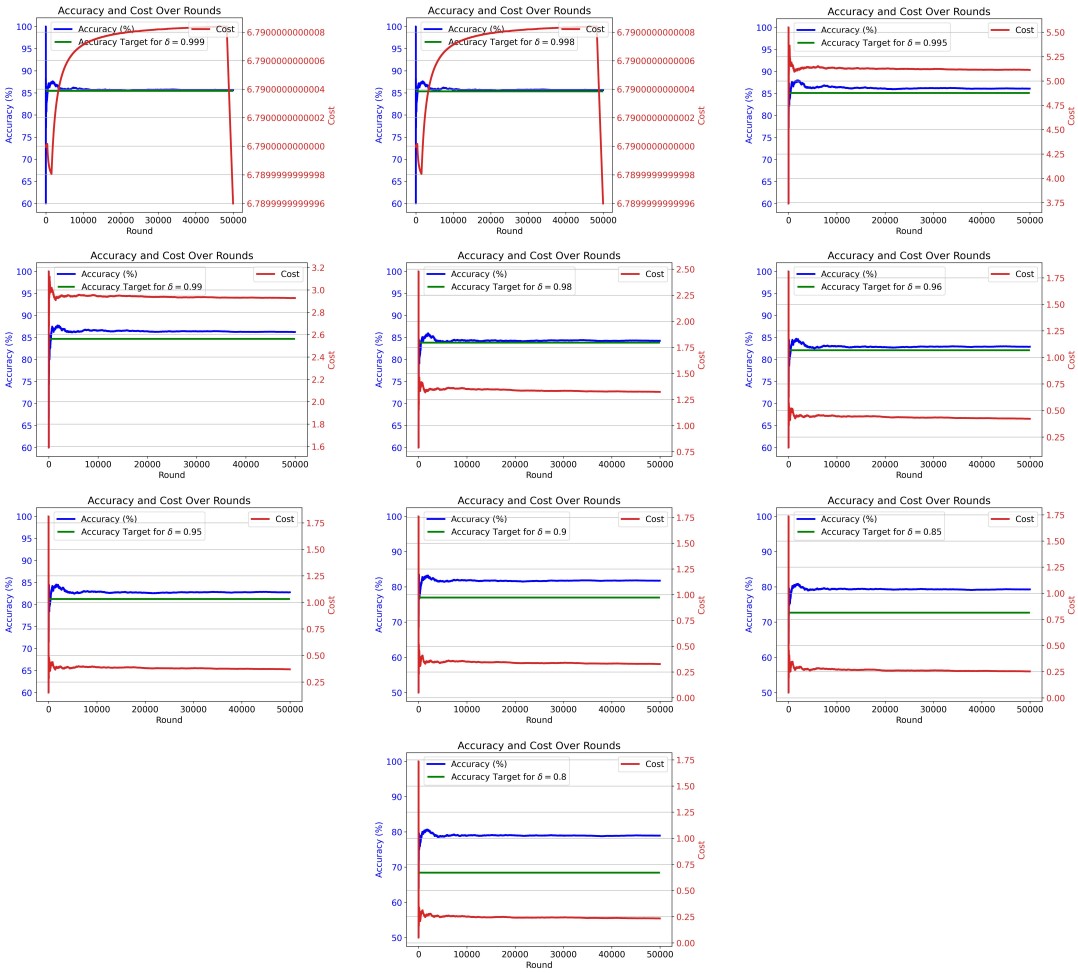

Figure 10: Cumulative average accuracy (blue) and cost (red) of CaMVo ($k_{\min} = 1$) on the AG News Classification Dataset across rounds for various confidence thresholds $\delta$. The green line marks each $\delta$-specific target accuracy.

CCaMVo differs from CaMVo by incorporating three additional components. The first is an online correlation estimator (Algorithm 3), which incrementally estimates pairwise correlations between LLM accuracies as new rounds of predictions arrive. Let $\mu_i$ and $\sigma_i$ denote the empirical mean and standard deviation of LLM $l_i$'s historical accuracy, respectively, and let $C$ denote the estimated correlation matrix across LLMs. For each LLM, the algorithm maintains running counts $n_i$, the number of times LLM $l_i$ has been selected up to that round. Further, for each pair $(i, j)$ of LLMs, $N_{ij}$, the co-occurrence counts that designate the number of times LLMs $l_i$ and $l_j$ have been selected at the same round; and $M_{ij}$, the accumulated deviation statistics, are maintained. At each round, it updates the univariate statistics $(\mu_i, \sigma_i)$ for selected LLMs and adjusts the terms $M_{ij}$ using Welford's numerically stable online formulas. Pairwise correlations are computed as $\rho_{ij} = \text{cov}_{ij}/(\sigma_i \sigma_j)$ where $\text{cov}_{ij} = M_{ij}/(N_{ij} - 1)$ and clipped to $[-1, 1]$ for numerical stability.

The second component (Algorithm 4) uses these estimated correlations to generate correlated LLM correctness samples via a Gaussian copula. Specifically, we first project the estimated correlation matrix $C_{\text{est}}$ to a positive semi-definite matrix $C_{\text{psd}}$ and draw $n$ samples $Z_1, \ldots, Z_n \sim \mathcal{N}(\mathbf{0}_K, C_{\text{psd}})$. Applying the standard normal CDF elementwise yields uniform samples $U_{ij} = \Phi(Z_{ij})$, which preserve the dependence structure of $C_{\text{psd}}$. Binary correctness samples are then obtained by thresholding each variable as $X_{ij} = \mathbb{1}\{U_{ij} < \mu_j\}$, ensuring marginal correctness probabilities of $\mu_j$. While the binary correlations differ slightly from $C_{\text{psd}}$ due to thresholding, this approximation offers a practical trade-off between accuracy and computational efficiency.

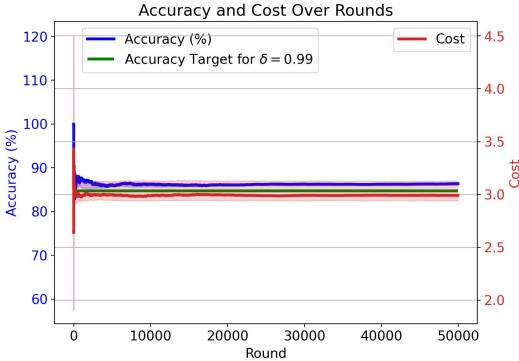

Figure 11: Mean (solid lines) and one-standard-deviation bands (shading) of CaMVo's cumulative average accuracy (blue) and cost (red) over 20 random shuffles using the AG News Classification Dataset ($\delta = 0.99$, $k_{\min} = 1$). The green line indicates the accuracy target of $95.14\%$ for $\delta = 0.995$.

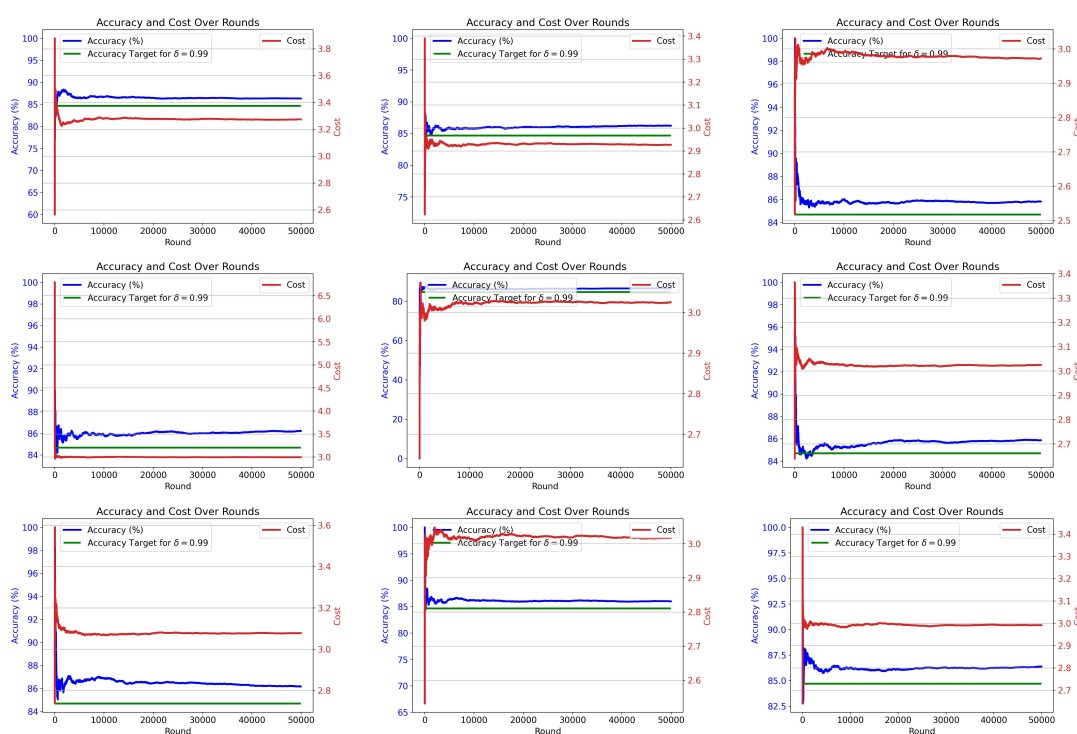

Figure 12: Cumulative average accuracy (blue) and cost (red) of CaMVo with $\delta = 0.99$, $k_{\min} = 1$ under nine different random permutations of the dataset. The green line marks each $\delta$-specific target accuracy.

Finally, CCaMVo estimates the confidence of each LLM subset by aggregating majority-vote correctness outcomes across the generated samples, averaging over simulated draws to approximate subset confidence. The remaining components of CCaMVo follow the same structure as CaMVo, preserving its efficiency while enabling robust confidence estimation under correlated LLM outputs.

We evaluated CCaMVo using the same experimental setup as CaMVo. On the MMLU dataset, Table 10 compares CaMVo and CCaMVo for $k_{\min} = 1$ across varying confidence thresholds $\delta$, while Table 11 shows the results for $k_{\min} = 3$. Similarly, for the IMDB dataset, Table 12 reports results for $k_{\min} = 1$, and Table 13 presents the corresponding results for $k_{\min} = 3$. For the AG News Classification Dataset, Table 14 reports results for $k_{\min} = 1$, and Table 15 presents the corresponding results for $k_{\min} = 3$.

---

**Algorithm 3** OnlineCorrelationEstimator

---

**Require:** Number of variables $K$
1: Initialize $n_k \leftarrow 0$, $\mu_k \leftarrow 0$, $\sigma_k \leftarrow 0$ for all $k = 1, \ldots, K$
2: Initialize $M_{ij} \leftarrow 0$, $N_{ij} \leftarrow 0$, and correlation matrix $C_{ij} \leftarrow \mathbb{I}[i = j]$
3: **function** UPDATE($\mathbf{r}, \mathcal{A}$)
$\qquad\qquad\qquad\qquad\qquad\qquad\qquad\qquad$ ▷ — Step 1: Update univariate statistics —
4: $\quad$ **for all** $i \in \mathcal{A}$ **do**
5: $\qquad$ old_$\mu_i \leftarrow \mu_i$
6: $\qquad$ $n_i \leftarrow n_i + 1$
7: $\qquad$ $\delta \leftarrow r_i - $ old_$\mu_i$
8: $\qquad$ $\mu_i \leftarrow$ old_$\mu_i + \delta/n_i$
9: $\qquad$ $M_{ii} \leftarrow M_{ii} + (r_i - \mu_i)(r_i - $ old_$\mu_i)$
10: $\qquad$ **if** $n_i > 1$ **then**
11: $\qquad\quad$ $\sigma_i \leftarrow \sqrt{M_{ii}/(n_i - 1)}$
12: $\qquad$ **else**
13: $\qquad\quad$ $\sigma_i \leftarrow 0$
14: $\qquad$ **end if**
15: $\quad$ **end for**
$\qquad\qquad\qquad\qquad\qquad\qquad$ ▷ — Step 2: Update pairwise covariance and correlation —
16: $\quad$ **for all** unordered pairs $(i, j)$ in $\mathcal{A}$ **do**
17: $\qquad$ $\delta_i \leftarrow r_i - \mu_i, \quad \delta_j \leftarrow r_j - \mu_j$
18: $\qquad$ $M_{ij} \leftarrow M_{ij} + \delta_i \delta_j$
19: $\qquad$ $M_{ji} \leftarrow M_{ij}$
20: $\qquad$ $N_{ij} \leftarrow N_{ij} + 1, \quad N_{ji} \leftarrow N_{ij}$
21: $\qquad$ **if** $N_{ij} > 1$ **then**
22: $\qquad\quad$ $\text{cov}_{ij} \leftarrow M_{ij}/(N_{ij} - 1)$
23: $\qquad\quad$ $\rho_{ij} \leftarrow \text{clip}(\text{cov}_{ij}/(\sigma_i \sigma_j), -1, 1)$
24: $\qquad\quad$ $C_{ij} \leftarrow C_{ji} \leftarrow \rho_{ij}$
25: $\qquad$ **end if**
26: $\quad$ **end for**
27: **end function**
28: **function** GETCORRELATIONMATRIX
29: $\quad$ **return** $C$
30: **end function**

---

---

**Algorithm 4** Sampling Correlated Binary Outcomes via Gaussian Copula

---

**Require:** Number of samples $n$, marginal probabilities $\mu \in \mathbb{R}^K$, estimated correlation matrix $C_{\text{est}} \in \mathbb{R}^{K \times K}$
1: $C_{\text{psd}} \leftarrow$ MAKEPSD($C_{\text{est}}$) $\qquad\qquad\qquad\qquad\qquad$ ▷ Project to nearest PSD matrix
2: $Z \leftarrow$ Sample $n$ vectors from $\mathcal{N}(\mathbf{0}_K, C_{\text{psd}})$
3: $U \leftarrow \Phi(Z)$ $\qquad\qquad\qquad\qquad\qquad\qquad$ ▷ Apply standard normal CDF elementwise
4: **for** $i = 1, \ldots, n$ **do**
5: $\quad$ **for** $j = 1, \ldots, K$ **do**
6: $\qquad$ $X_{ij} \leftarrow \mathbb{1}\{U_{ij} < \mu_j\}$
7: $\quad$ **end for**
8: **end for**
9: **return** $X$

---

These results indicate that incorporating the copula-based simulation to model the correlation does not significantly improve performance on MMLU and yields a modest cost improvement on IMDB, where LLM outputs exhibit higher correlation. We attribute this to two factors: (1) LinUCB's use of input embedding vectors to predict LLM confidences already implicitly captures correlations among LLM outputs conditioned on the input context, and (2) estimation errors in the correlation matrix reduce the potential gains from explicitly modeling correlations.

# H Experiments on Sensitivity to Correlated LLM Outputs

We conduct a sensitivity study to evaluate the impact of correlation among LLM outputs. Given a target mean accuracy vector $\mu = (\mu_1, \ldots, \mu_K)$ and a target covariance matrix $C \in \mathbb{R}^{K \times K}$, we generate synthetic LLM labels using a Gaussian copula–based procedure (Algorithm 5). This approach ensures that the outputs of different LLMs are both *context-dependent* and *correlated* according to the specified $C$. By varying $C$, we analyze how increasing inter-model correlation affects the performance of CaMVo.

In this experiment setting, we denote the number of LLMs by $K$, the context dimension by $d$, and the number of rounds by $T$. Each LLM $l_i$ is associated with a target mean accuracy $\mu_i \in (0, 1)$. The context correlation parameter $\alpha \in [0, 1]$ controls the degree of shared dependence among LLMs, and $\sigma > 0$ denotes the scale of latent Gaussian noise.

We first transform the target reward correlation matrix $C$ into a valid latent Gaussian correlation matrix $R = [\rho_{ij}]$ that can reproduce the desired binary LLM outputs after thresholding. Specifically, for each pair of LLMs $(i, j)$, we find a latent correlation coefficient $\rho_{ij}$ such that

$$\text{Corr}\big[\mathbb{I}\{Z_i > \Phi^{-1}(1 - \mu_i)\}, \mathbb{I}\{Z_j > \Phi^{-1}(1 - \mu_j)\}\big] = C_{ij},$$

where $(Z_i, Z_j) \sim \mathcal{N}(0, \Sigma(\rho_{ij}))$ and $\Phi^{-1}$ is the inverse standard normal CDF. The resulting matrix $R = [\rho_{ij}]$ is then projected onto the nearest positive semi-definite matrix to ensure numerical stability. This step guarantees that the generated binary outputs exhibit empirical correlations close to $C$.

Each LLM $l_i$ is associated with a context-dependent accuracy weight vector $\theta_i \in \mathbb{R}^d$ defined as

$$\theta_i = \alpha \, \theta_{\text{shared}} + (1 - \alpha) \, \theta_{\text{unique},i},$$

where $\theta_{\text{shared}} \sim \mathcal{N}(0, I_d)$ captures shared structure among LLMs, and $\theta_{\text{unique},i} \sim \mathcal{N}(0, I_d)$ introduces model-specific variation.

At each round $t = 1, \ldots, T$, a context vector

$$x_t \sim \mathcal{N}(0, I_d)$$

is independently sampled, representing the input context of the data instance. A correlated Gaussian noise vector $\epsilon_t \sim \mathcal{N}(0, R)$ is drawn jointly across all LLMs. The latent correctness score for each LLM $l_i$ is then computed as

$$z_{t,i} = \theta_i^\top x_t + \sigma \, \epsilon_{t,i}.$$

The observed binary correctness outcome is given by

$$r_{t,i} = \mathbb{I}\{z_{t,i} > \Phi^{-1}(1 - \mu_i)\}.$$

When $r_{t,i} = 1$, it means the LLM produces the correct label . This data generation process produces context–output pairs whose expected accuracies ar close to the target $\mu$, while the inter-model dependencies exhibit empirical correlations close to $C$. Note that because of the additive term $\theta_i$, which encodes correlations with the context, this method cannot exactly reproduce the target accuracy and covariance matrix values. Although more sophisticated approaches could more precisely match the specified targets, we adopt this formulation for its computational simplicity and because our experiments focus primarily on analyzing the effects of varying correlation strength rather than achieving exact target statistics.

## H.1 Experiment Results

In the first experimental setup the diagonal entries of $C$ representing self-correlation are set to 1, i.e. $C_{ii} = 1, \forall i \in [K]$ and all off-diagonal entries are set to a parameter $C_{ij} = \gamma, \forall i \neq j$ that represents the pairwise correlation. We sweep $\gamma$ across a range of values to evaluate how varying correlation strengths impact performance. The results are summarized in Table 16, which reports the accuracy and cost of CaMVo under $k_{\min} = 1$ and $k_{\min} = 3$, alongside the accuracy of majority voting and the target accuracy, across different confidence thresholds $\delta$ and correlation parameters $C_i$. The synthetic data are generated with parameters $d = 5$, $\sigma = 0.5$, and $\alpha = 0.1$, while the costs and mean accuracies of the LLMs are taken from the MMLU experiments, given by $c = [0.05, 1.1, 0.59, 2.5, 3, 1.1, 0.8]$ and $\mu = [0.6801, 0.8482, 0.8170, 0.8358, 0.8565, 0.8592, 0.6409]$. Note that the cost of majority

---

**Algorithm 5** Gaussian Copula–based Correlated Data Generation

---

**Require:** Number of rounds $T$, number of LLMs $K$, context dimension $d$, target mean rewards $\{\mu_i\}_{i=1}^K$, target correlation matrix $C \in \mathbb{R}^{K \times K}$, context correlation parameter $\alpha$, noise scale $\sigma$
**Ensure:** Contexts $\{x_t\}_{t=1}^T$, binary rewards $\{r_{t,i}\}_{t,i}$

1: **function** GENERATEDATA($T, K, d, \mu, C, \alpha, \sigma$)
2:     Compute latent Gaussian correlation matrix $R \leftarrow$ CALIBRATECOPULA($\mu, C$)
3:     Draw $\theta_{\text{shared}} \sim \mathcal{N}(0, I_d)$
4:     **for** $a = 1, \ldots, K$ **do**
5:         Draw $\theta_{\text{unique},a} \sim \mathcal{N}(0, I_d)$
6:         $\theta_a \leftarrow \alpha\, \theta_{\text{shared}} + (1 - \alpha)\, \theta_{\text{unique},a}$
7:     **end for**
8:     **for** $t = 1, \ldots, T$ **do**
9:         Sample context $x_t \sim \mathcal{N}(0, I_d)$
10:        Sample $\epsilon_t \sim \mathcal{N}(0, R)$
11:        **for** $a = 1, \ldots, K$ **do**
12:            $z_{t,a} \leftarrow \theta_a^\top x_t + \sigma\, \epsilon_{t,a}$
13:            $r_{t,a} \leftarrow \mathbb{I}\{z_{t,a} > \Phi^{-1}(1 - \mu_a)\}$
14:        **end for**
15:    **end for**
16:    **return** $\{(x_t, r_t)\}_{t=1}^T$
17: **end function**

18: **function** CALIBRATECOPULA($\mu, C$)
19:     **for** each pair $(i, j)$ with $i < j$ **do**
20:         Find $\rho_{ij}$ such that

$$\text{Corr}\big[\mathbb{I}\{Z_i > \Phi^{-1}(1 - \mu_i)\}, \mathbb{I}\{Z_j > \Phi^{-1}(1 - \mu_j)\}\big] = C_{ij}$$

     where $(Z_i, Z_j) \sim \mathcal{N}(0, \Sigma(\rho_{ij}))$
21:     **end for**
22:     Construct $R = [\rho_{ij}]$ and project to nearest PSD matrix
23:     **return** $R$
24: **end function**

---

voting is fixed at \$9.14 per million tokens across all settings. Tables 17 and 18 present the results for $\alpha = 0.2$ and $\alpha = 0.4$, respectively.

The results indicate that the accuracy of CaMVo generally decreases as $C_i$ increases; a similar trend is observed for majority voting. Importantly, CaMVo is able to maintain the target accuracy across all tested correlation levels, demonstrating its effectiveness even in settings with correlated LLM outputs.

In the second experimental setup, we use the correlation matrix

$$C = \begin{bmatrix} 1.000 & 0.445 & 0.147 & 0.423 & 0.211 & 0.306 & 0.320 \\ 0.445 & 1.000 & 0.373 & 0.448 & 0.242 & 0.420 & 0.317 \\ 0.147 & 0.373 & 1.000 & 0.265 & 0.172 & 0.118 & 0.142 \\ 0.423 & 0.448 & 0.265 & 1.000 & 0.301 & 0.436 & 0.325 \\ 0.211 & 0.242 & 0.172 & 0.301 & 1.000 & 0.158 & 0.180 \\ 0.306 & 0.420 & 0.118 & 0.436 & 0.158 & 1.000 & 0.388 \\ 0.320 & 0.317 & 0.142 & 0.325 & 0.180 & 0.388 & 1.000 \end{bmatrix} \tag{22}$$

so that, unlike the first experiment, the correlations between different LLMs are heterogeneous. We sweep the confidence threshold $\delta$ and the parameter $\alpha$ to evaluate how the strength of correlation between LLM outputs and the context affects accuracy. The results are summarized in Table 19, which reports the accuracy and cost of CaMVo for $k_{\min} = 1$ and $k_{\min} = 3$, alongside the accuracy of majority voting and the target accuracy, across varying $\delta$ and $\alpha$. Across all experiments, $d = 5$ and $\sigma = 0.5$, while the costs and mean accuracies of the LLMs are taken from the MMLU dataset as in the previous experiment. The cost of majority voting remains fixed at \$9.14 per million tokens.

As expected, the accuracy of CaMVo increases with larger $\alpha$, as higher correlation with the context allows LLM outputs to better reflect the arm rewards. In contrast, the accuracy of majority voting decreases with increasing $\alpha$, since it increases correlations among LLMs through the context. Crucially, CaMVo consistently maintains the target accuracy across all instances, demonstrating its robustness even under correlated LLM outputs.

| $\delta$ | label | Precision, $k_{min}=1$ | Recall, $k_{min}=1$ | F1 Score, $k_{min}=1$ | Precision, $k_{min}=3$ | Recall, $k_{min}=3$ | F1 Score, $k_{min}=3$ |
|---|---|---|---|---|---|---|---|
| 0.99 | 0 | 0.87 | 0.86 | 0.87 | 0.87 | 0.86 | 0.87 |
| 0.99 | 1 | 0.87 | 0.89 | 0.88 | 0.87 | 0.89 | 0.88 |
| 0.99 | 2 | 0.88 | 0.9 | 0.89 | 0.88 | 0.9 | 0.89 |
| 0.99 | 3 | 0.91 | 0.89 | 0.9 | 0.91 | 0.89 | 0.9 |
| 0.98 | 0 | 0.88 | 0.86 | 0.87 | 0.88 | 0.86 | 0.87 |
| 0.98 | 1 | 0.88 | 0.89 | 0.88 | 0.88 | 0.89 | 0.88 |
| 0.98 | 2 | 0.88 | 0.9 | 0.89 | 0.88 | 0.9 | 0.89 |
| 0.98 | 3 | 0.91 | 0.9 | 0.9 | 0.91 | 0.9 | 0.9 |
| 0.975 | 0 | 0.88 | 0.85 | 0.87 | 0.88 | 0.85 | 0.87 |
| 0.975 | 1 | 0.87 | 0.89 | 0.88 | 0.87 | 0.89 | 0.88 |
| 0.975 | 2 | 0.88 | 0.9 | 0.89 | 0.88 | 0.9 | 0.89 |
| 0.975 | 3 | 0.91 | 0.9 | 0.9 | 0.91 | 0.9 | 0.9 |
| 0.97 | 0 | 0.88 | 0.85 | 0.86 | 0.88 | 0.85 | 0.86 |
| 0.97 | 1 | 0.87 | 0.89 | 0.88 | 0.87 | 0.89 | 0.88 |
| 0.97 | 2 | 0.88 | 0.9 | 0.89 | 0.88 | 0.9 | 0.89 |
| 0.97 | 3 | 0.91 | 0.9 | 0.9 | 0.91 | 0.9 | 0.9 |
| 0.965 | 0 | 0.89 | 0.85 | 0.87 | 0.89 | 0.85 | 0.87 |
| 0.965 | 1 | 0.86 | 0.88 | 0.87 | 0.86 | 0.88 | 0.87 |
| 0.965 | 2 | 0.88 | 0.9 | 0.89 | 0.88 | 0.9 | 0.89 |
| 0.965 | 3 | 0.91 | 0.9 | 0.9 | 0.91 | 0.9 | 0.9 |
| 0.96 | 0 | 0.88 | 0.85 | 0.86 | 0.88 | 0.85 | 0.86 |
| 0.96 | 1 | 0.86 | 0.88 | 0.87 | 0.86 | 0.88 | 0.87 |
| 0.96 | 2 | 0.87 | 0.89 | 0.88 | 0.87 | 0.89 | 0.88 |
| 0.96 | 3 | 0.9 | 0.89 | 0.9 | 0.9 | 0.89 | 0.9 |
| 0.955 | 0 | 0.87 | 0.84 | 0.86 | 0.87 | 0.84 | 0.86 |
| 0.955 | 1 | 0.86 | 0.88 | 0.87 | 0.86 | 0.88 | 0.87 |
| 0.955 | 2 | 0.87 | 0.88 | 0.88 | 0.87 | 0.88 | 0.88 |
| 0.955 | 3 | 0.89 | 0.89 | 0.89 | 0.89 | 0.89 | 0.89 |
| 0.95 | 0 | 0.87 | 0.84 | 0.85 | 0.87 | 0.84 | 0.85 |
| 0.95 | 1 | 0.85 | 0.87 | 0.86 | 0.85 | 0.87 | 0.86 |
| 0.95 | 2 | 0.87 | 0.87 | 0.87 | 0.87 | 0.87 | 0.87 |
| 0.95 | 3 | 0.89 | 0.89 | 0.89 | 0.89 | 0.89 | 0.89 |
| 0.9 | 0 | 0.85 | 0.82 | 0.84 | 0.85 | 0.82 | 0.84 |
| 0.9 | 1 | 0.84 | 0.85 | 0.84 | 0.84 | 0.85 | 0.84 |
| 0.9 | 2 | 0.85 | 0.85 | 0.85 | 0.85 | 0.85 | 0.85 |
| 0.9 | 3 | 0.86 | 0.87 | 0.87 | 0.86 | 0.87 | 0.87 |
| 0.85 | 0 | 0.84 | 0.82 | 0.83 | 0.84 | 0.82 | 0.83 |
| 0.85 | 1 | 0.81 | 0.86 | 0.84 | 0.81 | 0.86 | 0.84 |
| 0.85 | 2 | 0.85 | 0.84 | 0.84 | 0.85 | 0.84 | 0.84 |
| 0.85 | 3 | 0.87 | 0.85 | 0.86 | 0.87 | 0.85 | 0.86 |
| 0.8 | 0 | 0.8 | 0.82 | 0.81 | 0.8 | 0.82 | 0.81 |
| 0.8 | 1 | 0.76 | 0.87 | 0.81 | 0.76 | 0.87 | 0.81 |
| 0.8 | 2 | 0.85 | 0.8 | 0.82 | 0.85 | 0.8 | 0.82 |
| 0.8 | 3 | 0.88 | 0.8 | 0.84 | 0.88 | 0.8 | 0.84 |
| 0.75 | 0 | 0.59 | 0.73 | 0.65 | 0.59 | 0.73 | 0.65 |
| 0.75 | 1 | 0.71 | 0.65 | 0.68 | 0.71 | 0.65 | 0.68 |
| 0.75 | 2 | 0.76 | 0.63 | 0.69 | 0.76 | 0.63 | 0.69 |
| 0.75 | 3 | 0.72 | 0.74 | 0.73 | 0.72 | 0.74 | 0.73 |
| 0.7 | 0 | 0.58 | 0.73 | 0.65 | 0.58 | 0.73 | 0.65 |
| 0.7 | 1 | 0.71 | 0.64 | 0.67 | 0.71 | 0.64 | 0.67 |
| 0.7 | 2 | 0.75 | 0.63 | 0.68 | 0.75 | 0.63 | 0.68 |
| 0.7 | 3 | 0.71 | 0.74 | 0.73 | 0.71 | 0.74 | 0.73 |

Table 5: Precision, recall, and F1 scores of CaMVo with $k_{min}=1$ and $k_{min}=3$ across different confidence thresholds $\delta$ for each label category on the MMLU dataset.

|  | label | Precision | Recall | F1 Score |
|---|---|---|---|---|
| Maj. voting | 0 | 0.87 | 0.86 | 0.86 |
| Maj. voting | 1 | 0.87 | 0.88 | 0.88 |
| Maj. voting | 2 | 0.88 | 0.89 | 0.89 |
| Maj. voting | 3 | 0.91 | 0.89 | 0.9 |

Table 6: Precision, recall and F1 score of majority voting over each output category on the MMLU dataset.

| $\delta$ | label | Precision ($k_{min} = 1$) | Recall ($k_{min} = 1$) | F1 Score ($k_{min} = 1$) | Precision ($k_{min} = 3$) | Recall ($k_{min} = 3$) | F1 Score ($k_{min} = 3$) |
|---|---|---|---|---|---|---|---|
| 0.999 | 0 | 0.96 | 0.96 | 0.96 | 0.96 | 0.96 | 0.96 |
| 0.999 | 1 | 0.96 | 0.96 | 0.96 | 0.96 | 0.96 | 0.96 |
| 0.998 | 0 | 0.95 | 0.95 | 0.95 | 0.95 | 0.95 | 0.95 |
| 0.998 | 1 | 0.95 | 0.95 | 0.95 | 0.95 | 0.95 | 0.95 |
| 0.997 | 0 | 0.96 | 0.95 | 0.95 | 0.96 | 0.95 | 0.95 |
| 0.997 | 1 | 0.95 | 0.96 | 0.95 | 0.95 | 0.96 | 0.95 |
| 0.995 | 0 | 0.96 | 0.95 | 0.95 | 0.96 | 0.95 | 0.95 |
| 0.995 | 1 | 0.95 | 0.96 | 0.95 | 0.95 | 0.96 | 0.95 |
| 0.99 | 0 | 0.95 | 0.95 | 0.95 | 0.95 | 0.95 | 0.95 |
| 0.99 | 1 | 0.95 | 0.95 | 0.95 | 0.95 | 0.95 | 0.95 |
| 0.985 | 0 | 0.94 | 0.95 | 0.95 | 0.94 | 0.95 | 0.95 |
| 0.985 | 1 | 0.95 | 0.94 | 0.95 | 0.95 | 0.94 | 0.95 |
| 0.98 | 0 | 0.94 | 0.95 | 0.95 | 0.94 | 0.95 | 0.95 |
| 0.98 | 1 | 0.95 | 0.94 | 0.95 | 0.95 | 0.94 | 0.95 |
| 0.97 | 0 | 0.94 | 0.95 | 0.95 | 0.94 | 0.95 | 0.95 |
| 0.97 | 1 | 0.95 | 0.94 | 0.95 | 0.95 | 0.94 | 0.95 |
| 0.96 | 0 | 0.94 | 0.94 | 0.94 | 0.94 | 0.94 | 0.94 |
| 0.96 | 1 | 0.94 | 0.94 | 0.94 | 0.94 | 0.94 | 0.94 |
| 0.95 | 0 | 0.94 | 0.95 | 0.94 | 0.94 | 0.95 | 0.94 |
| 0.95 | 1 | 0.95 | 0.94 | 0.94 | 0.95 | 0.94 | 0.94 |
| 0.9 | 0 | 0.94 | 0.94 | 0.94 | 0.94 | 0.94 | 0.94 |
| 0.9 | 1 | 0.94 | 0.94 | 0.94 | 0.94 | 0.94 | 0.94 |
| MJV | 0 | 0.96 | 0.96 | 0.96 | | | |
| MJV | 1 | 0.96 | 0.96 | 0.96 | | | |

Table 7: Precision, recall, and F1 scores of CaMVo with $k_{min} = 1$ and $k_{min} = 3$ across different confidence thresholds $\delta$ for each output label on the IMDB dataset. Note that the last two rows report the corresponding precision, recall, and F1 scores obtained with majority voting.

| LLM / Method | Accuracy (%) | Cost |
|---|---|---|
| o3-mini | 87.16 | 1.10 |
| llama-3.3-70b | 86.35 | 0.59 |
| o1-mini | 85.77 | 1.10 |
| claude-3-7-sonnet | 85.27 | 3.00 |
| claude-3-5-haiku | 84.71 | 0.80 |
| gpt-4o-mini | 82.63 | 0.15 |
| llama-3.1-8b | 77.85 | 0.05 |
| Majority Vote | 85.55 | 6.79 |
| Baseline Method | 85.68 | 6.79 |

Table 8: Accuracy and cost of individual LLMs and baseline ensemble methods on the AG News Classification Dataset.

| CaMVo $\delta$ | Target Acc. (%) | Acc. (%) $k_{\min} = 1$ | Cost $k_{\min} = 1$ | Acc. (%) $k_{\min} = 3$ | Cost $k_{\min} = 3$ |
|---|---|---|---|---|---|
| 0.999 | 85.59 | 85.98 | 6.82 | 85.98 | 6.82 |
| 0.998 | 85.51 | 85.98 | 6.82 | 85.98 | 6.82 |
| 0.995 | 85.25 | 86.36 | 5.24 | 86.36 | 5.24 |
| 0.99 | 84.82 | 86.18 | 2.98 | 86.18 | 2.98 |
| 0.98 | 83.97 | 84.72 | 1.39 | 84.72 | 1.39 |
| 0.96 | 82.25 | 83.38 | 0.52 | 83.85 | 0.99 |
| 0.95 | 81.40 | 83.18 | 0.40 | 83.77 | 0.96 |
| 0.9 | 77.11 | 80.66 | 0.29 | 83.75 | 0.88 |
| 0.85 | 72.83 | 79.62 | 0.26 | 83.74 | 0.87 |
| 0.80 | 68.54 | 79.34 | 0.23 | 83.73 | 0.87 |

Table 9: Accuracy and cost of CaMVo on the AG News Classification Dataset under varying confidence thresholds $\delta$ and $k_{\min} \in \{1, 3\}$. For reference, the cost of the baseline method is \$6.79 per million tokens.

| $\delta$ | Target Acc. (%) | CaMVo Acc. (%) | CaMVo Cost | CCaMVo Acc. (%) | CCaMVo Cost |
|---|---|---|---|---|---|
| 0.99 | 87.30 | 88.47 | 9.14 | 88.55 | 9.14 |
| 0.98 | 86.42 | 88.59 | 8.57 | 88.53 | 8.72 |
| 0.975 | 85.98 | 88.49 | 7.80 | 88.41 | 8.14 |
| 0.97 | 85.53 | 88.35 | 6.67 | 88.25 | 7.10 |
| 0.965 | 85.09 | 88.27 | 5.66 | 88.07 | 4.88 |
| 0.96 | 84.65 | 87.98 | 4.74 | 86.96 | 3.07 |
| 0.955 | 84.21 | 87.40 | 3.38 | 86.84 | 2.58 |
| 0.95 | 83.77 | 86.82 | 2.76 | 86.33 | 2.32 |
| 0.90 | 79.36 | 84.88 | 1.19 | 82.29 | 0.76 |
| 0.85 | 74.95 | 84.41 | 1.03 | 81.72 | 0.67 |
| 0.80 | 70.54 | 82.12 | 0.70 | 80.09 | 0.60 |
| 0.75 | 66.14 | 68.80 | 0.16 | 68.30 | 0.14 |
| 0.70 | 61.73 | 68.38 | 0.14 | 68.30 | 0.14 |

Table 10: Accuracy and cost of CaMVo and Correlated CaMVo (CCaMVo) on the MMLU dataset under varying confidence thresholds $\delta$ and $k_{\min} = 1$. For reference, the cost of the baseline method is \$9.14 per million tokens.

| $\delta$ | Target Acc. (%) | CaMVo Acc. (%) | CaMVo Cost | CCaMVo Acc. (%) | CCaMVo Cost |
|---|---|---|---|---|---|
| 0.99 | 87.30 | 88.47 | 9.14 | 88.55 | 9.14 |
| 0.98 | 86.42 | 88.59 | 8.57 | 88.50 | 8.73 |
| 0.975 | 85.98 | 88.49 | 7.80 | 88.43 | 8.16 |
| 0.97 | 85.53 | 88.33 | 6.67 | 88.33 | 6.90 |
| 0.965 | 85.09 | 88.27 | 5.66 | 88.29 | 5.21 |
| 0.96 | 84.65 | 88.03 | 4.74 | 86.40 | 3.73 |
| 0.955 | 84.21 | 87.01 | 3.36 | 87.08 | 2.69 |
| 0.95 | 83.77 | 87.01 | 2.96 | 86.42 | 2.45 |
| 0.90 | 79.36 | 84.80 | 1.81 | 84.17 | 1.77 |
| 0.85 | 74.95 | 82.14 | 1.58 | 81.45 | 1.52 |
| 0.80 | 70.54 | 81.32 | 1.51 | 81.23 | 1.50 |
| 0.75 | 66.14 | 81.24 | 1.50 | 81.18 | 1.50 |
| 0.70 | 61.73 | 81.22 | 1.50 | 81.17 | 1.50 |

Table 11: Accuracy and cost of CaMVo and Correlated CaMVo (CCaMVo) on the MMLU dataset under varying confidence thresholds $\delta$ and $k_{\min} = 3$. For reference, the cost of the baseline method is \$9.14 per million tokens.

| $\delta$ | Target Acc. (%) | CaMVo Acc. (%) | CaMVo Cost | CCaMVo Acc. (%) | CCaMVo Cost |
|---|---|---|---|---|---|
| 0.999 | 95.52 | 95.59 | 6.15 | 95.43 | 3.61 |
| 0.998 | 95.43 | 95.43 | 4.03 | 95.28 | 2.29 |
| 0.997 | 95.33 | 95.45 | 2.83 | 95.10 | 1.32 |
| 0.995 | 95.14 | 95.25 | 2.06 | 95.09 | 0.99 |
| 0.99 | 94.66 | 95.10 | 1.09 | 95.09 | 0.89 |
| 0.985 | 94.20 | 94.69 | 0.34 | 95.05 | 0.82 |
| 0.98 | 93.71 | 94.69 | 0.31 | 94.78 | 0.38 |
| 0.97 | 92.75 | 94.56 | 0.22 | 94.68 | 0.28 |
| 0.96 | 91.80 | 94.21 | 0.13 | 94.18 | 0.20 |
| 0.95 | 90.84 | 94.28 | 0.14 | 94.61 | 0.26 |
| 0.9 | 86.06 | 94.24 | 0.10 | 94.11 | 0.15 |

Table 12: Accuracy and cost of CaMVo and Correlated CaMVo (CCaMVo) on the IMDB dataset under varying confidence thresholds $\delta$ and $k_{\min} = 1$. For reference, the cost of the baseline method is \$6.29 per million tokens.

| $\delta$ | Target Acc. (%) | CaMVo Acc. (%) | CaMVo Cost | CCaMVo Acc. (%) | CCaMVo Cost |
|---|---|---|---|---|---|
| 0.999 | 95.52 | 95.59 | 6.15 | 95.46 | 3.61 |
| 0.998 | 95.43 | 95.43 | 4.03 | 95.29 | 2.21 |
| 0.997 | 95.33 | 95.45 | 2.83 | 95.10 | 1.21 |
| 0.995 | 95.14 | 95.25 | 2.06 | 95.10 | 0.99 |
| 0.99 | 94.66 | 95.12 | 0.99 | 95.07 | 0.88 |
| 0.985 | 94.20 | 95.06 | 0.84 | 95.07 | 0.85 |
| 0.98 | 93.71 | 95.07 | 0.83 | 95.08 | 0.83 |
| 0.97 | 92.75 | 95.07 | 0.82 | 95.07 | 0.83 |
| 0.96 | 91.80 | 95.06 | 0.81 | 95.07 | 0.82 |
| 0.95 | 90.84 | 95.07 | 0.81 | 95.06 | 0.82 |
| 0.9 | 86.06 | 95.06 | 0.81 | 95.06 | 0.81 |

Table 13: Accuracy and cost of CaMVo and Correlated CaMVo (CCaMVo) on the IMDB dataset under varying confidence thresholds $\delta$ and $k_{\min} = 3$. For reference, the cost of the baseline method is \$6.29 per million tokens.

| $\delta$ | Target Acc. (%) | CaMVo Acc. (%) | CaMVo Cost | CCaMVo Acc. (%) | CCaMVo Cost |
|---|---|---|---|---|---|
| 0.999 | 85.59 | 85.98 | 6.82 | 85.98 | 6.82 |
| 0.998 | 85.51 | 85.98 | 6.82 | 85.98 | 6.81 |
| 0.995 | 85.25 | 86.36 | 5.24 | 86.10 | 5.12 |
| 0.99 | 84.82 | 86.18 | 2.98 | 85.82 | 2.75 |
| 0.98 | 83.97 | 84.72 | 1.39 | 83.94 | 1.13 |
| 0.96 | 82.25 | 83.38 | 0.52 | 83.3 | 0.51 |
| 0.95 | 81.40 | 83.18 | 0.40 | 83.18 | 0.44 |
| 0.9 | 77.11 | 80.66 | 0.29 | 81.5 | 0.33 |
| 0.85 | 72.83 | 79.62 | 0.26 | 79.66 | 0.26 |
| 0.80 | 68.54 | 79.34 | 0.23 | 79.26 | 0.24 |

Table 14: Accuracy and cost of CaMVo and Correlated CaMVo (CCaMVo) on the AG News Classification Dataset under varying confidence thresholds $\delta$ and $k_{\min} = 1$. For reference, the cost of the baseline method is \$6.79 per million tokens.

| $\delta$ | Target Acc. (%) | CaMVo Acc. (%) | CaMVo Cost | CCaMVo Acc. (%) | CCaMVo Cost |
|---|---|---|---|---|---|
| 0.999 | 85.59 | 85.98 | 6.82 | 85.98 | 6.82 |
| 0.998 | 85.51 | 85.98 | 6.82 | 85.98 | 6.81 |
| 0.995 | 85.25 | 86.36 | 5.24 | 86.14 | 5.08 |
| 0.99 | 84.82 | 86.18 | 2.98 | 85.67 | 2.29 |
| 0.98 | 83.97 | 84.72 | 1.39 | 83.94 | 1.10 |
| 0.96 | 82.25 | 83.85 | 0.99 | 83.89 | 1.01 |
| 0.95 | 81.40 | 83.77 | 0.96 | 83.75 | 0.93 |
| 0.9 | 77.11 | 83.75 | 0.88 | 83.75 | 0.89 |
| 0.85 | 72.83 | 83.74 | 0.87 | 83.72 | 0.87 |
| 0.80 | 68.54 | 83.73 | 0.87 | 83.72 | 0.87 |

Table 15: Accuracy and cost of CaMVo and Correlated CaMVo (CCaMVo) on the AG News Classification Dataset under varying confidence thresholds $\delta$ and $k_{\min} = 3$. For reference, the cost of the baseline method is \$6.79 per million tokens.

| $\delta$ | $\gamma$ | Maj. vote Acc. (%) | Target Acc. (%) | CaMVo ($k_{min} = 1$) Acc. (%) | CaMVo ($k_{min} = 1$) Cost | CaMVo ($k_{min} = 3$) Acc. (%) | CaMVo ($k_{min} = 3$) Cost |
|---|---|---|---|---|---|---|---|
| 0.96 | 0.1 | 93.36 | 89.63 | 93.35 | 9.14 | 93.40 | 9.14 |
| 0.96 | 0.2 | 92.84 | 89.13 | 92.83 | 9.14 | 92.80 | 9.14 |
| 0.96 | 0.3 | 92.30 | 88.61 | 92.30 | 9.14 | 92.30 | 9.14 |
| 0.96 | 0.4 | 91.32 | 87.67 | 91.32 | 9.14 | 91.30 | 9.14 |
| 0.96 | 0.5 | 90.99 | 87.35 | 90.99 | 9.14 | 91.00 | 9.14 |
| 0.96 | 0.6 | 90.63 | 87.00 | 90.63 | 9.14 | 90.60 | 9.14 |
| 0.96 | 0.7 | 89.84 | 86.25 | 89.84 | 9.14 | 89.80 | 9.14 |
| 0.96 | 0.8 | 89.16 | 85.59 | 89.16 | 9.14 | 89.20 | 9.14 |
| 0.95 | 0.0 | 93.85 | 89.16 | 93.85 | 9.14 | 93.90 | 9.14 |
| 0.95 | 0.1 | 93.36 | 88.69 | 93.35 | 9.14 | 93.40 | 9.14 |
| 0.95 | 0.2 | 92.84 | 88.20 | 92.83 | 9.14 | 92.80 | 9.14 |
| 0.95 | 0.3 | 92.30 | 87.69 | 92.30 | 9.14 | 92.30 | 9.14 |
| 0.95 | 0.4 | 91.32 | 86.76 | 91.32 | 9.14 | 91.30 | 9.14 |
| 0.95 | 0.5 | 90.99 | 86.44 | 90.99 | 9.14 | 91.00 | 9.14 |
| 0.95 | 0.6 | 90.63 | 86.09 | 90.63 | 9.14 | 90.60 | 9.14 |
| 0.95 | 0.7 | 89.84 | 85.35 | 89.84 | 9.14 | 89.80 | 9.14 |
| 0.95 | 0.8 | 89.16 | 84.70 | 89.16 | 9.14 | 89.20 | 9.14 |
| 0.93 | 0.0 | 93.85 | 87.28 | 93.87 | 7.59 | 93.90 | 7.59 |
| 0.93 | 0.1 | 93.36 | 86.83 | 92.74 | 7.20 | 92.70 | 7.20 |
| 0.93 | 0.2 | 92.84 | 86.34 | 90.76 | 5.68 | 90.80 | 5.68 |
| 0.93 | 0.3 | 92.30 | 85.84 | 91.52 | 6.54 | 91.50 | 6.54 |
| 0.93 | 0.4 | 91.32 | 84.93 | 90.31 | 6.03 | 90.30 | 6.03 |
| 0.93 | 0.5 | 90.99 | 84.62 | 89.56 | 5.43 | 89.60 | 5.43 |
| 0.93 | 0.6 | 90.63 | 84.28 | 89.35 | 5.11 | 89.40 | 5.11 |
| 0.93 | 0.7 | 89.84 | 83.55 | 88.68 | 4.57 | 88.70 | 4.57 |
| 0.93 | 0.8 | 89.16 | 82.92 | 88.23 | 4.35 | 88.20 | 4.35 |
| 0.90 | 0.0 | 93.85 | 84.47 | 86.98 | 2.23 | 86.70 | 2.66 |
| 0.90 | 0.1 | 93.36 | 84.03 | 86.79 | 2.23 | 86.20 | 2.57 |
| 0.90 | 0.2 | 92.84 | 83.56 | 86.84 | 2.03 | 87.50 | 2.63 |
| 0.90 | 0.3 | 92.30 | 83.07 | 85.61 | 3.53 | 87.80 | 3.78 |
| 0.90 | 0.4 | 91.32 | 82.19 | 85.68 | 1.85 | 84.50 | 2.29 |
| 0.90 | 0.5 | 90.99 | 81.89 | 85.50 | 1.69 | 84.10 | 2.15 |
| 0.90 | 0.6 | 90.63 | 81.56 | 85.23 | 1.66 | 83.70 | 2.16 |
| 0.90 | 0.7 | 89.84 | 80.86 | 84.69 | 1.49 | 83.10 | 2.02 |
| 0.90 | 0.8 | 89.16 | 80.24 | 84.64 | 1.47 | 82.80 | 2.01 |
| 0.80 | 0.0 | 93.85 | 75.08 | 84.56 | 1.13 | 80.10 | 1.55 |
| 0.80 | 0.1 | 93.36 | 74.69 | 84.35 | 1.16 | 80.80 | 1.56 |
| 0.80 | 0.2 | 92.84 | 74.27 | 84.58 | 1.12 | 80.00 | 1.52 |
| 0.80 | 0.3 | 92.30 | 73.84 | 84.42 | 1.14 | 79.10 | 1.51 |
| 0.80 | 0.4 | 91.32 | 73.06 | 84.54 | 1.14 | 79.10 | 1.52 |
| 0.80 | 0.5 | 90.99 | 72.79 | 84.68 | 1.13 | 78.70 | 1.49 |
| 0.80 | 0.6 | 90.63 | 72.50 | 84.74 | 1.15 | 78.60 | 1.50 |
| 0.80 | 0.7 | 89.84 | 71.87 | 83.05 | 0.92 | 78.20 | 1.47 |
| 0.80 | 0.8 | 89.16 | 71.33 | 80.90 | 0.64 | 78.30 | 1.48 |

Table 16: Accuracy and cost of CaMVo with $k_{min} = 1$ and $k_{min} = 3$, alongside the accuracy of majority voting and the target accuracy, under varying confidence thresholds $\delta$ and correlation parameters $C_i$ in the synthetic correlated data. The synthetic data are generated with parameters $d = 5$, $\sigma = 0.5$, and $\alpha = 0.1$. Note that the cost of majority voting is fixed at \$9.14 per million tokens for all settings.

| $\delta$ | $\gamma$ | Maj. vote Acc. (%) | Target Acc. (%) | CaMVo ($k_{min} = 1$) Acc. (%) | CaMVo ($k_{min} = 1$) Cost | CaMVo ($k_{min} = 3$) Acc. (%) | CaMVo ($k_{min} = 3$) Cost |
|---|---|---|---|---|---|---|---|
| 0.96 | 0.0 | 93.95 | 90.19 | 93.95 | 9.14 | 93.95 | 9.14 |
| 0.96 | 0.1 | 93.43 | 89.69 | 93.43 | 9.14 | 93.43 | 9.14 |
| 0.96 | 0.2 | 92.86 | 89.15 | 92.86 | 9.14 | 92.86 | 9.14 |
| 0.96 | 0.3 | 92.28 | 88.59 | 92.28 | 9.14 | 92.28 | 9.14 |
| 0.96 | 0.4 | 91.57 | 87.91 | 91.57 | 9.14 | 91.57 | 9.14 |
| 0.96 | 0.5 | 91.07 | 87.43 | 91.07 | 9.14 | 91.07 | 9.14 |
| 0.96 | 0.6 | 90.65 | 87.02 | 90.65 | 9.14 | 90.65 | 9.14 |
| 0.96 | 0.7 | 90.06 | 86.46 | 90.06 | 9.14 | 90.06 | 9.14 |
| 0.96 | 0.8 | 89.41 | 85.83 | 89.41 | 9.14 | 89.41 | 9.14 |
| 0.95 | 0.0 | 93.95 | 89.25 | 96.00 | 8.54 | 96.00 | 8.54 |
| 0.95 | 0.1 | 93.43 | 88.75 | 95.84 | 8.51 | 95.84 | 8.51 |
| 0.95 | 0.2 | 92.86 | 88.22 | 95.48 | 8.50 | 95.48 | 8.50 |
| 0.95 | 0.3 | 92.28 | 87.66 | 94.95 | 8.54 | 94.95 | 8.54 |
| 0.95 | 0.4 | 91.57 | 86.99 | 94.54 | 8.52 | 94.54 | 8.52 |
| 0.95 | 0.5 | 91.07 | 86.51 | 94.25 | 8.50 | 94.25 | 8.50 |
| 0.95 | 0.6 | 90.65 | 86.12 | 91.65 | 7.82 | 91.65 | 7.82 |
| 0.95 | 0.7 | 90.06 | 85.56 | 89.90 | 6.97 | 89.90 | 6.97 |
| 0.95 | 0.8 | 89.41 | 84.94 | 89.76 | 6.60 | 89.76 | 6.60 |
| 0.90 | 0.0 | 93.95 | 84.55 | 86.55 | 1.71 | 85.40 | 2.24 |
| 0.90 | 0.1 | 93.43 | 84.08 | 86.44 | 3.09 | 90.51 | 3.27 |
| 0.90 | 0.2 | 92.86 | 83.58 | 86.32 | 1.68 | 84.94 | 2.14 |
| 0.90 | 0.3 | 92.28 | 83.05 | 87.63 | 2.20 | 88.51 | 2.91 |
| 0.90 | 0.4 | 91.57 | 82.42 | 86.06 | 1.64 | 84.53 | 2.12 |
| 0.90 | 0.5 | 91.07 | 81.96 | 85.72 | 1.52 | 84.98 | 2.02 |
| 0.90 | 0.6 | 90.65 | 81.58 | 84.09 | 1.98 | 84.09 | 1.98 |
| 0.90 | 0.7 | 90.06 | 81.06 | 85.31 | 1.44 | 83.69 | 1.99 |
| 0.90 | 0.8 | 89.41 | 80.47 | 85.55 | 1.36 | 82.90 | 1.95 |
| 0.80 | 0.0 | 93.95 | 75.16 | 85.32 | 1.13 | 80.11 | 1.51 |
| 0.80 | 0.1 | 93.43 | 74.74 | 85.33 | 1.15 | 80.10 | 1.53 |
| 0.80 | 0.2 | 92.86 | 74.29 | 82.66 | 0.73 | 79.91 | 1.50 |
| 0.80 | 0.3 | 92.28 | 73.82 | 85.24 | 1.14 | 78.78 | 1.49 |
| 0.80 | 0.4 | 91.57 | 73.26 | 85.27 | 1.13 | 78.67 | 1.48 |
| 0.80 | 0.5 | 91.07 | 72.85 | 85.30 | 1.13 | 78.76 | 1.47 |
| 0.80 | 0.6 | 90.65 | 72.52 | 84.12 | 0.98 | 78.58 | 1.48 |
| 0.80 | 0.7 | 90.06 | 72.05 | 81.77 | 0.63 | 78.40 | 1.48 |
| 0.80 | 0.8 | 89.41 | 71.53 | 85.45 | 1.13 | 77.93 | 1.47 |

Table 17: Accuracy and cost of CaMVo with $k_{min} = 1$ and $k_{min} = 3$, alongside the accuracy of majority voting and the target accuracy, under varying confidence thresholds $\delta$ and correlation parameters $C_i$ in the synthetic correlated data. The synthetic data are generated with parameters $d = 5$, $\sigma = 0.5$, and $\alpha = 0.2$. Note that the cost of majority voting is fixed at \$9.14 per million tokens for all settings.

| $\delta$ | $\gamma$ | Maj. vote Acc. (%) | Target Acc. (%) | CaMVo ($k_{min} = 1$) Acc. (%) | CaMVo ($k_{min} = 1$) Cost | CaMVo ($k_{min} = 3$) Acc. (%) | CaMVo ($k_{min} = 3$) Cost |
|---|---|---|---|---|---|---|---|
| 0.96 | 0.1 | 91.83 | 88.16 | 96.69 | 6.62 | 96.69 | 6.62 |
| 0.96 | 0.2 | 91.54 | 87.88 | 96.77 | 6.54 | 96.77 | 6.54 |
| 0.96 | 0.3 | 90.75 | 87.12 | 96.18 | 6.64 | 96.18 | 6.64 |
| 0.96 | 0.4 | 90.28 | 86.67 | 95.58 | 6.46 | 95.58 | 6.46 |
| 0.96 | 0.5 | 89.61 | 86.02 | 95.68 | 6.31 | 95.68 | 6.31 |
| 0.96 | 0.6 | 89.27 | 85.70 | 95.28 | 6.24 | 95.28 | 6.24 |
| 0.96 | 0.7 | 88.79 | 85.24 | 89.15 | 4.99 | 89.15 | 4.99 |
| 0.96 | 0.8 | 88.32 | 84.79 | 88.07 | 4.45 | 88.07 | 4.45 |
| 0.95 | 0 | 92.45 | 87.83 | 97.50 | 5.69 | 97.50 | 5.69 |
| 0.95 | 0.1 | 91.83 | 87.24 | 96.55 | 5.52 | 96.55 | 5.52 |
| 0.95 | 0.2 | 91.54 | 86.96 | 97.07 | 5.51 | 97.07 | 5.51 |
| 0.95 | 0.3 | 90.75 | 86.21 | 96.47 | 5.56 | 96.47 | 5.56 |
| 0.95 | 0.4 | 90.28 | 85.77 | 95.85 | 5.41 | 95.85 | 5.41 |
| 0.95 | 0.5 | 89.61 | 85.13 | 95.33 | 5.14 | 95.33 | 5.14 |
| 0.95 | 0.6 | 89.27 | 84.81 | 91.03 | 4.74 | 91.03 | 4.74 |
| 0.95 | 0.7 | 88.79 | 84.35 | 88.68 | 3.46 | 88.68 | 3.46 |
| 0.95 | 0.8 | 88.32 | 83.90 | 94.52 | 5.00 | 94.52 | 5.00 |
| 0.93 | 0 | 92.45 | 85.98 | 92.72 | 3.56 | 92.72 | 3.56 |
| 0.93 | 0.1 | 91.83 | 85.40 | 91.07 | 3.48 | 91.07 | 3.48 |
| 0.93 | 0.2 | 91.54 | 85.13 | 90.44 | 3.23 | 90.44 | 3.23 |
| 0.93 | 0.3 | 90.75 | 84.39 | 90.23 | 3.34 | 90.23 | 3.34 |
| 0.93 | 0.4 | 90.28 | 83.96 | 90.12 | 3.21 | 90.12 | 3.21 |
| 0.93 | 0.5 | 89.61 | 83.33 | 89.63 | 3.26 | 89.63 | 3.26 |
| 0.93 | 0.6 | 89.27 | 83.02 | 86.25 | 1.95 | 86.25 | 1.95 |
| 0.93 | 0.7 | 88.79 | 82.57 | 86.35 | 1.88 | 86.35 | 1.88 |
| 0.93 | 0.8 | 88.32 | 82.14 | 86.45 | 1.66 | 86.45 | 1.66 |
| 0.9 | 0 | 92.45 | 83.20 | 86.80 | 1.42 | 86.80 | 1.42 |
| 0.9 | 0.1 | 91.83 | 82.65 | 85.45 | 2.03 | 85.45 | 2.03 |
| 0.9 | 0.2 | 91.54 | 82.38 | 86.63 | 1.52 | 86.63 | 1.52 |
| 0.9 | 0.3 | 90.75 | 81.67 | 86.78 | 1.57 | 86.78 | 1.57 |
| 0.9 | 0.4 | 90.28 | 81.26 | 83.97 | 1.90 | 83.97 | 1.90 |
| 0.9 | 0.5 | 89.61 | 80.65 | 83.75 | 1.44 | 83.75 | 1.44 |
| 0.9 | 0.6 | 89.27 | 80.35 | 82.89 | 1.08 | 82.89 | 1.08 |
| 0.9 | 0.7 | 88.79 | 79.91 | 86.36 | 1.28 | 86.36 | 1.28 |
| 0.9 | 0.8 | 88.32 | 79.49 | 82.69 | 0.92 | 82.69 | 0.92 |
| 0.8 | 0 | 92.45 | 73.96 | 82.53 | 0.61 | 82.53 | 0.61 |
| 0.8 | 0.1 | 91.83 | 73.46 | 86.15 | 1.13 | 86.15 | 1.13 |
| 0.8 | 0.2 | 91.54 | 73.23 | 85.71 | 1.10 | 85.71 | 1.10 |
| 0.8 | 0.3 | 90.75 | 72.60 | 85.88 | 1.14 | 85.88 | 1.14 |
| 0.8 | 0.4 | 90.28 | 72.23 | 82.57 | 0.64 | 82.57 | 0.64 |
| 0.8 | 0.5 | 89.61 | 71.69 | 86.01 | 1.13 | 86.01 | 1.13 |
| 0.8 | 0.6 | 89.27 | 71.42 | 82.70 | 0.63 | 82.70 | 0.63 |
| 0.8 | 0.7 | 88.79 | 71.03 | 82.85 | 0.64 | 82.85 | 0.64 |
| 0.8 | 0.8 | 88.32 | 70.65 | 82.71 | 0.62 | 82.71 | 0.62 |

Table 18: Accuracy and cost of CaMVo with $k_{min} = 1$ and $k_{min} = 3$, alongside the accuracy of majority voting and the target accuracy, under varying confidence thresholds $\delta$ and correlation parameters $C_i$ in the synthetic correlated data. The synthetic data are generated with parameters $d = 5$, $\sigma = 0.5$, and $\alpha = 0.4$. Note that the cost of majority voting is fixed at \$9.14 per million tokens for all settings.

| $\alpha$ | $\delta$ | Maj. vote Acc. (%) | Target Acc. (%) | CaMVo ($k_{min} = 1$) Acc. (%) | CaMVo ($k_{min} = 1$) Cost | CaMVo ($k_{min} = 3$) Acc. (%) | CaMVo ($k_{min} = 3$) Cost |
|---|---|---|---|---|---|---|---|
| 0.1 | 0.96 | 92.21 | 88.53 | 92.21 | 9.14 | 92.21 | 9.14 |
| 0.1 | 0.95 | 92.21 | 87.60 | 92.21 | 9.14 | 92.21 | 9.14 |
| 0.1 | 0.93 | 92.21 | 85.76 | 92.41 | 6.97 | 92.41 | 6.97 |
| 0.1 | 0.9 | 92.21 | 82.99 | 85.82 | 1.81 | 85.82 | 1.81 |
| 0.1 | 0.85 | 92.21 | 78.38 | 84.61 | 1.21 | 84.61 | 1.21 |
| 0.1 | 0.8 | 92.21 | 73.77 | 80.73 | 0.66 | 80.73 | 0.66 |
| 0.2 | 0.96 | 92.36 | 88.67 | 92.36 | 9.14 | 92.36 | 9.14 |
| 0.2 | 0.95 | 92.36 | 87.75 | 94.94 | 8.53 | 94.94 | 8.53 |
| 0.2 | 0.93 | 92.36 | 85.90 | 91.12 | 5.10 | 91.12 | 5.10 |
| 0.2 | 0.9 | 92.36 | 83.13 | 85.69 | 1.45 | 85.69 | 1.45 |
| 0.2 | 0.85 | 92.36 | 78.51 | 85.35 | 1.18 | 85.35 | 1.18 |
| 0.2 | 0.8 | 92.36 | 73.89 | 84.93 | 1.10 | 84.93 | 1.10 |
| 0.4 | 0.96 | 91.23 | 87.58 | 93.59 | 6.70 | 93.59 | 6.70 |
| 0.4 | 0.95 | 91.23 | 86.67 | 96.61 | 5.49 | 96.61 | 5.49 |
| 0.4 | 0.93 | 91.23 | 84.85 | 90.10 | 3.03 | 90.10 | 3.03 |
| 0.4 | 0.9 | 91.23 | 82.11 | 84.71 | 1.91 | 84.71 | 1.91 |
| 0.4 | 0.85 | 91.23 | 77.55 | 82.96 | 0.69 | 82.96 | 0.69 |
| 0.4 | 0.8 | 91.23 | 72.99 | 82.90 | 0.63 | 82.90 | 0.63 |

Table 19: Accuracy and cost of CaMVo with $k_{min} = 1$ and $k_{min} = 3$, alongside the accuracy of majority voting and the target accuracy, $\delta$ and $\alpha$ values for synthetic data generated using the correlation matrix in Eq. (22). The synthetic data are generated with parameters $d = 5$, and $\sigma = 0.5$. Note that the cost of majority voting is fixed at \$9.14 per million tokens for all settings.

