# OpenReview forum: "Cost-aware LLM-based Online Dataset Annotation"
_NeurIPS.cc/2025/Conference — NeurIPS 2025 spotlight_

### Official Review · Reviewer_U74S · 2025-06-20

**Clarity:** 3
**Significance:** 3
**Originality:** 3
**Rating:** 4
**Confidence:** 3

**Summary:**

The paper introduces Cost-aware Majority Voting (CaMVo), an online framework that labels data with large language-model (LLM) ensembles while minimizing inference cost. CaMVo first uses a LinUCB-style contextual bandit to compute, for every candidate LLM, a lower-confidence bound (LCB) on its probability of producing the correct label for the current instance. A Bayesian Beta-mixture then converts that LCB into a posterior correctness estimate, and a knapsack-style “Oracle’’ picks the cheapest subset of models whose weighted-vote confidence exceeds a user-specified threshold and a minimum vote count. If no subset passes the test, all models are queried. Experiments on MMLU and IMDB show that CaMVo matches or surpasses full-ensemble majority-vote accuracy while cutting cost by about 40 % on MMLU and about 50 % on IMDB under some settings.

**Questions:**

1. How do you plan to test and, if necessary, correct for the violation of the conditional-independence assumption that underlies Lemma 2.1 and the Oracle’s confidence bound? (e.g., measuring pair-wise correlations)
2. How will you extend the method beyond classification tasks?

**Ethical Concerns:**

["NO or VERY MINOR ethics concerns only"]

**Final Justification:**

The authors provided clear and detailed responses that resolved the main concerns. They clarified the definition of relative accuracy, justified the use of the Beta prior and Laplace regularization, and explained the subset selection complexity with both exact and approximate strategies. The correlation sensitivity study using a Gaussian copula showed that CaMVo remains effective even when LLM outputs are correlated. The empirical results are strong, with CaMVo often matching or exceeding the Pareto frontier while significantly reducing cost. Responses to reviewer questions were thoughtful and demonstrated practical awareness. While evaluation is limited to two datasets and correlation modeling is not fully integrated, the core method is technically sound, well-motivated, and impactful. I recommend acceptance.

**Limitations:**

Yes.

**Paper Formatting Concerns:**

No.

**Quality:**

3

**Strengths And Weaknesses:**

**Quality**: the paper thoroughly specifies CaMVo, including pseudocode and all hyper-parameters. The experiments were conducted on two benchmarks, seven off-the-shelf LLMs, and ablations over $\omega$ and $k_\min$ demonstrate consistent cost savings at equal (or better) accuracy. However, the theory and subset-confidence bound assume conditional independence of LLM outputs, a condition the authors admit is violated on the IMDB sentiment task and can degrade performance.

**Clarity**: writing is generally clear, with a step-by-step derivation of the bound and well-annotated figures showing cost–accuracy trade-offs.

**Significance**: this paper addresses a practical bottleneck: escalating API cost of ensemble labelling at a time when LLM annotation pipelines are widely adopted. It demonstrates sizeable savings without ground-truth labels or offline training. The paper can benefit more from analysis on harder structured-output problems or real-time latency. Practical impact thus remains to be validated beyond the chosen domains.

**Originality**: this paper proposes the first fully online cost-aware subset-of-LLMs selection scheme that jointly adapts both voting weights and the subset size, combining contextual bandits with a Bayesian confidence model. However, it is conceptually related to prior bandit-based router and query-budget work mentioned in the related work section.

---

> ### Author Rebuttal · Authors · 2025-07-31
>
> Dear Reviewer,
>
> Thank you for the comments. Our responses to the reviewer's questions are below.
>
> **Q1:** Lemma 2.1 can be extended to remove the conditional-independence assumption using Bahadur’s model (R.R. Bahadur, “On classification based on responses to n dichotomous items,” 1961) characterizes the joint distribution of $n$ binary variables using their marginal probabilities $p_i$, second-order dependence terms known as Bahadur parameters, and higher-order interaction terms. This framework allows the computation of majority vote confidence by summing the probabilities of all outcomes where the majority label is correct. However, the lack of a closed-form solution and the rapid growth in computational complexity with increasing $n$ limit its practical applicability. We will include a note on this in the discussion of Lemma 2.1 in the final version of the paper.
>
> Due to this computational intractability, we implemented a practical Monte Carlo simulation approach to evaluate majority vote confidence given LLM accuracies and their correlations. Using a Gaussian copula, we generate $m$ binary outcomes (correct/incorrect) for each LLM that capture both the marginal accuracies and correlations. From these simulated outcomes, we compute $m$ majority vote results weighted by the LLM weights. Average accuracy over these instances provides an empirical confidence estimate for the majority voting scheme.
>
> Applying this method on the MMLU dataset with CaMVo ($\delta=0.96$, $k_{\min}=1$) yields an accuracy of 86.96\% at a cost of 3.07, exceeding the target accuracy of 84.65\%. For comparison, the original CaMVo results are 87.98\% accuracy at 4.74 cost ($\delta=0.96$), and 86.82\% accuracy at 2.76 cost ($\delta=0.95$). On the IMDB dataset, the simulation with $\delta=0.99$ and $k_{\min}=1$ produces 95.09\% accuracy at a cost of 0.83, surpassing the target accuracy of 94.66\%. The original CaMVo results are 95.10\% accuracy at 1.09 cost ($\delta=0.99$), and 94.69\% accuracy at 0.34 cost ($\delta=0.985$).
>
> These results indicate that incorporating the copula-based simulation does not significantly improve performance on MMLU and yields a modest cost improvement on IMDB, where LLM outputs exhibit higher correlation. We attribute this to two factors: (1) LinUCB’s use of input embedding vectors to predict LLM confidences already implicitly captures correlations among LLM outputs conditioned on the input context, and (2) estimation errors in the correlation matrix reduce the potential gains from explicitly modeling correlations.
>
> We plan to extend this study across all tested $\delta$ and $k_{\min}$ values in our simulations and include these results in the final version of the paper.
>
>
> **Q2:** Our method can be naturally extended beyond classification tasks in several ways. For regression, LLM outputs can be aggregated via a weighted average or median, with weights updated based on how closely each LLM’s output aligns with the aggregate. For ranking tasks, each LLM can assign scores to items, and a weighted combination of these scores would produce a final ranking; weights can then be adjusted according to agreement of the LLM's ranking generated from its scores with the aggregated ranking. These extensions involve only minor modifications to the aggregation and weight update steps, making them straightforward to integrate into our existing framework. We will highlight these as directions for future work in the final version of the paper.

---

> > ### Comment · Reviewer_U74S · 2025-08-05
> >
> > Thank you for the detailed rebuttals and clarifications. They were helpful in understanding the design choices behind CaMVo, I appreciate the additional discussion on independence violations and possible extensions beyond classification tasks.
> >
> > At this stage, I lean toward accepting the paper based on its strong theoretical grounding, practical relevance, and thorough responses to reviewers. While the evaluation is still limited to two datasets and correlation modeling is not fully integrated into the method, the proposed framework addresses a clear and timely problem in cost-aware LLM annotation.
> >
> > I will also review your discussions with other reviewers to see if remaining concerns, particularly regarding broader empirical validation and correlation-aware modeling, are further addressed. My score may change based on that discussion.

---

### Official Review · Reviewer_Ld4S · 2025-06-29

**Clarity:** 3
**Significance:** 3
**Originality:** 3
**Rating:** 5
**Confidence:** 3

**Summary:**

This paper introduces a new algorithm called Cost-aware Majority Voting (CaMVo) for efficient and accurate dataset labeling using LLMs. Traditional approaches to LLM-based annotation often rely on majority voting across multiple models to improve label reliability, but this incurs high computational costs. CaMVo addresses this by adaptively selecting a subset of LLMs for each data instance, balancing cost and confidence without requiring pre-training or ground-truth labels. It uses a LinUCB-based contextual bandit algorithm combined with a Bayesian confidence estimator to determine which models to query, aiming to meet a user-defined accuracy threshold while minimizing cost.

The core innovation of CaMVo lies in its online, cost-aware selection mechanism. For each data instance, it computes contextual embeddings and estimates a lower bound on each LLM’s probability of producing a correct label. These estimates are used to select the smallest-cost subset of models whose aggregated confidence exceeds a specified threshold. The algorithm also incorporates regularization using Laplace smoothing to prevent overfitting and uses a weighted majority voting scheme where weights reflect both historical and expected instance-specific performance. This dynamic adaptation allows CaMVo to operate effectively without offline training or labeled data.

Empirical evaluations on two datasets, MMLU and IMDB, demonstrate CaMVo’s effectiveness. On MMLU, CaMVo achieves higher or comparable accuracy to full majority voting while reducing costs by around 40%. On the IMDB dataset, where individual LLMs already perform well, CaMVo still maintains high accuracy with comparable or lower cost. These results highlight CaMVo’s ability to adapt to different task complexities and cost constraints, suggesting it may be a practical tool for scalable annotation.

CaMVo is the first fully online method that jointly adapts both the subset of queried models and their voting weights on a per-instance basis. By leveraging contextual embeddings and probabilistic confidence estimation, CaMVo achieves strong performance without relying on labeled data or pre-training. Its flexibility and efficiency make it well-suited for dynamic labeling environments, and its design opens avenues for further research into more robust and adaptive ensemble strategies.

**Questions:**

How did you decide on the individual models, especially since they didn’t necessarily exhibit comparable performance levels?

How often did each LLM give an incorrectly formatted response? Did this complicate the evaluations?

In practice, how do you choose a confidence threshold?

How might results compare with human annotation? Is there a good indication that LLMs approximate human annotation for these types of tasks?

How did you choose the embedding model? What informed your decision?

How could CaMVo be extended to incorporate human in the loop annotations?

**Ethical Concerns:**

["NO or VERY MINOR ethics concerns only"]

**Final Justification:**

I thank the authors for their additional comments and answers to my questions. My final score remains the same.

**Limitations:**

The paper discusses limitations, particularly the assumption of independence among LLM outputs. In practice, LLMs may produce correlated outputs, which can reduce the benefits of majority voting and lead to suboptimal performance. This issue is especially evident in simpler tasks like IMDB sentiment analysis, where ensemble methods may not outperform the best single model. The authors suggest future work could involve modeling inter-model correlations and incorporating diversity-aware selection strategies.

**Quality:**

3

**Strengths And Weaknesses:**

Strengths:
-	The proposed algorithm seems very practical for real-world annotation scenarios.
-	The authors take cost into account and design the selection of the subset of LLMs to reduce cost, again making this very practical for real world applications.
-	The work is based on a strong theoretical foundation.
-	The two empirical examples give an intuitive understanding of both the strengths and weaknesses of the approach.

Weaknesses:
-	As noted by the authors, the assumption of independence in LLM outputs is likely problematic.
-	The algorithm was only tested on two datasets; additional empirical evidence for the efficacy of the approach is needed.
-	Accuracy was the only evaluation metric considered.
-	No guidance or insight around how to select a confidence threshold is provided.

---

> ### Author Rebuttal · Authors · 2025-07-31
>
> Dear Reviewer,
>
> Thank you for the comments. Our responses to the reviewer's comments are below.
>
> ## Responses to Questions:
>
> **Q1:** In selecting the individual models, we prioritized those commonly used in related work while also aiming to ensure diversity in both model architecture and per-token cost. Since our primary objective is cost optimization, it is important to span a wide range of cost levels to enable meaningful trade-offs.
>
> Additionally, we included different versions of similar LLMs with varying parameter sizes (e.g., llama-3.1 and llama-3.3) to allow CaMVo to learn cost-aware selection strategies. The intent is to allow CaMVo to favor cheaper, smaller models on simpler tasks, and to choose the larger, more expensive models when the input task is more complex.
>
>
> **Q2:**  On the MMLU dataset, more than 20\% of the LLM responses were incorrectly formatted. For example, when the expected output was the answer choice ‘A’, common variations included responses like “A) 5”, “5”, “The answer is A”, or full explanations preceding the answer. Similarly, on the IMDB dataset, LLMs frequently generated summaries of user comments before stating their classification decision.
>
> To handle these inconsistencies, we post-processed the outputs using keyword matching and other simple heuristics to extract the intended answer. In fewer than 2\% of cases, the LLMs failed to produce a usable response (i.e. by answering a question in the movie review in the IMDB dataset or providing an answer that is not in the answer choices). In such cases, we were able to obtain valid responses by requerying the LLM.
>
>
> **Q3:**  In practice, the confidence threshold can be set to reflect a desired target accuracy. If no specific accuracy requirement is given, the threshold can instead be adapted dynamically to meet a target cost budget or tuned over time as feedback on the output labels becomes available. This flexibility allows the method to operate under varying application constraints, whether accuracy-driven, cost-driven, or feedback-guided.
>
>
> **Q4:**  Yes, there is a growing body of work showing that LLMs can approximate, and in some cases even surpass human annotators across a wide range of tasks, including the classification tasks considered in our work. For example, Gilardi et al. (“ChatGPT Outperforms Crowd-Workers for Text-Annotation Tasks”) demonstrate that LLMs can not only match but often exceed the performance of crowd-sourced human annotations.
>
>
>
> **Q5:**  Since CaMVo is designed for online use, low-latency embedding generation is crucial. We selected SBERT over BERT because SBERT is significantly faster for producing sentence embeddings. To inform our choice in choosing an SBERT model, we referred to the accuracy and speed trade-off comparisons provided on the official Sentence Transformers (SBERT) library website. We chose the all-MiniLM-L6-v2 model as it offers a strong balance between efficiency and performance, making it well-suited for our setting.
>
>
>
> **Q6:** In the current implementation of CaMVo, we query all LLMs when no subset achieves confidence above the threshold $\delta$. For example, with $\delta=0.96$ and $k_{\min}=1$ on the MMLU dataset, this occurs an average of 597 times across 14,042 data instances. To extend this mechanism, an additional threshold $\delta_{\text{human}} < \delta$ can be introduced: if the highest subset confidence falls between $\delta_{\text{human}}$ and $\delta$, we query all LLMs; if it falls below $\delta_{\text{human}}$, the instance can be deferred to a human annotator. This extension provides a principled way to incorporate human in the loop annotations for low-confidence labels.
>
>
> ## Responses to Weaknesses:
>
> **W1:** Please see our response to the limitations section.
>
> **W2:** We will further extend our simulation results to an additional dataset and include the results in the final version of the paper, and we have conducted the following synthetic experiment to analyze the sensitivity of our method to correlated LLM outputs, and will also include its results in the final version of the paper. Using the true accuracies of the LLMs, we apply the Gaussian copula approach to generate synthetic labels that preserve the individual accuracies while introducing controlled correlations via a specified correlation matrix. The diagonal entries are set to 1, and off-diagonal entries are set to a shared correlation coefficient $C_i$, which we vary across experiments. The results from these experiments show that while CaMVo’s performance declines with increasing correlation, the performance of majority voting also degrades. Importantly, CaMVo maintains its target accuracy except under very high correlation levels. See our response W5 to Reviewer fBz7 for additional details.
>
>
> **W3:** We will include additional evaluation metrics such as precision, recall, and F1 score in the final version of the paper to provide a more comprehensive assessment of performance.
>
> **W4:** Please refer to our response Q3.
>
>
> ## Responses to Limitations
>
> **L1:** To overcome the assumption of independence, we have implemented a practical Monte Carlo simulation approach to evaluate majority vote confidence given LLM accuracies and their correlations. Using a Gaussian copula, we generate $m$ binary outcomes (correct/incorrect) for each LLM that capture both the marginal accuracies and correlations. From these simulated outcomes, we compute $m$ majority vote results weighted by the LLM weights. Average accuracy over these instances provides an empirical confidence estimate for the majority voting scheme.
>
> Applying this method on the MMLU dataset with CaMVo ($\delta=0.96$, $k_{\min}=1$) yields an accuracy of 86.96\% at a cost of 3.07, exceeding the target accuracy of 84.65\%. For comparison, the original CaMVo results are 87.98\% accuracy at 4.74 cost ($\delta=0.96$), and 86.82\% accuracy at 2.76 cost ($\delta=0.95$). On the IMDB dataset, the simulation with $\delta=0.99$ and $k_{\min}=1$ produces 95.09\% accuracy at a cost of 0.83, surpassing the target accuracy of 94.66\%. The original CaMVo results are 95.10\% accuracy at 1.09 cost ($\delta=0.99$), and 94.69\% accuracy at 0.34 cost ($\delta=0.985$).
>
> These results indicate that incorporating the copula-based simulation does not significantly improve performance on MMLU and yields a modest cost improvement on IMDB, where LLM outputs exhibit higher correlation. We attribute this to two factors: (1) LinUCB’s use of input embedding vectors to predict LLM confidences already implicitly captures correlations among LLM outputs conditioned on the input context, and (2) estimation errors in the correlation matrix reduce the potential gains from explicitly modeling correlations.
>
> We plan to extend this study across all tested $\delta$ and $k_{\min}$ values in our simulations and include these results in the final version of the paper.
>
> Additionally, Lemma 2.1 can be extended to remove the conditional-independence assumption using Bahadur’s model (R.R. Bahadur, “On classification based on responses to n dichotomous items,” 1961) characterizes the joint distribution of $n$ binary variables using their marginal probabilities $p_i$, second-order dependence terms known as Bahadur parameters, and higher-order interaction terms. This framework allows the computation of majority vote confidence by summing the probabilities of all outcomes where the majority label is correct. However, the lack of a closed-form solution and the rapid growth in computational complexity with increasing $n$ limit its practical applicability. We will include a note on this in the discussion of Lemma 2.1 in the final version of the paper.

---

> > ### Comment · Reviewer_Ld4S · 2025-08-04
> >
> > Thank you for the responses and for extending your work. My rating (accept) remains the same.

---

### Official Review · Reviewer_ZiN8 · 2025-07-01

**Clarity:** 3
**Significance:** 2
**Originality:** 3
**Rating:** 5
**Confidence:** 4

**Summary:**

The core of this paper lies in exploring the methods that if it is possible to dramatically reduce the cost of dataset annotation by adaptively selecting a small, optimized subset of LLMs for each specific data instance, while achieving an accuracy comparable or even superior to using a full, expensive ensemble of all available LLMs.

**Questions:**

1. Given CaMVo’s focus on _online_ cost efficiency (§1.1, Table 1), how might it integrate task-specific fine-tuning (e.g., LoRA adapters) without violating the zero-shot constraint? Could dynamically tuned LLMs alter the cost-accuracy trade-off?
2. In correlated-error regimes (e.g., IMDB), does the Beta-mixture estimator’s performance degrade? Would modeling inter-model dependencies (e.g., via copula-based confidence aggregation) enhance robustness?

**Ethical Concerns:**

["NO or VERY MINOR ethics concerns only"]

**Final Justification:**

I recommend Acceptance. The authors' rebuttal successfully resolved my primary concerns regarding the NP-hard complexity and the independence assumption. Their proposed greedy algorithm and copula-based simulation are practical solutions that make the work significantly more robust. With these key issues addressed, the paper is technically sound, and my initial reservations are gone.

**Limitations:**

1. **Independence Assumption**: As noted in §5, correlated LLM errors reduce ensemble gains, particularly in low-diversity tasks (e.g., IMDB). This remains an open challenge.

**Paper Formatting Concerns:**

1. Algorithm 1’s update step for Est_i (§3) should explicitly reference Appendix B’s Beta-parameter methods

**Quality:**

2

**Strengths And Weaknesses:**

**Strengths**

1. The authors combine two well-established concepts -- contextual bandits and Bayesian estimation, the design is clean and makes intuitive sense: use the bandit to learn which models are good at which types of questions, and use the Bayesian estimator to quantify the underlying reliability of each model.
2. The online approach make the method more practical for many
3. The use of Pareto frontier plots provides a clear, visual benchmark of how the system performs against a theoretical "perfect" selector.

**Weakness**

1. The NP-hard nature of subset selection (Eq. 4) warrants deeper discussion. While the paper mentions approximation via Beta-CDF, the absence of complexity analysis or runtime metrics obscures practical deployability.
2. The independence assumption (§2, Lemma 2.1) undermines performance when LLM outputs correlate strongly (e.g., IMDB in §4.2). While acknowledged in limitations, preliminary mitigation strategies (e.g., covariance-aware confidence bounds) could be explored.

---

> ### Author Rebuttal · Authors · 2025-07-31
>
> Dear Reviewer,
>
> Thank you for the comments. Our responses to the reviewer's comments are below.
>
> ## Responses to Questions:
>
> **Q1:** We propose the following implementation strategy to integrate task-specific fine-tuning. Initially, we will start with zero-shot LLMs to label data. As more data is labeled, we can fine-tune the LLMs using the accumulated labels. However, blindly fine-tuning on all generated labels risks incorporating incorrect annotations, which can degrade model performance and lead to error propagation. To mitigate this, we introduce a threshold $\delta_u$ and update the LLMs only with labels whose confidence exceeds $\delta_u$.
>
> Since CaMVo already selects subsets with confidence above a threshold $\delta$, it will not necessarily choose subsets with confidence above $\delta_u$. To enable this, we can use  $\delta_u$ as the update threshold for a user-defined fraction of the data. Additionally, as more data is collected, we can recalculate label confidences and include them for fine-tuning once they surpass $\delta_u$. To control cost, fine-tuning can be performed in batches once a sufficient number of high-confidence labels is accumulated.
>
> To account for changes in model behavior due to fine-tuning when estimating model parameters such as empirical accuracy, we can use discounted updates on model parameters, assigning higher weight to recent observations.
>
> Finally, since fine-tuning may increase correlation among LLMs, it may be beneficial to keep the original zero-shot LLMs available for selection in the subset. In such cases, we can introduce a constraint to avoid simultaneously selecting both the fine-tuned and zero-shot versions of the same model, as they are likely to be highly correlated.
>
>
> **Q2:** Yes, since the Beta estimator does not model correlations between LLMs, its performance will degrade in regimes with correlated LLM outputs. To overcome this degradation in performance, and also to remove the assumption of independence, we have implemented a practical Monte Carlo simulation approach to evaluate majority vote confidence given LLM accuracies and their correlations. Using a Gaussian copula, we generate $m$ binary outcomes (correct/incorrect) for each LLM that capture both the marginal accuracies and correlations. From these simulated outcomes, we compute $m$ majority vote results weighted by the LLM weights. Average accuracy over these instances provides an empirical confidence estimate for the majority voting scheme.
>
> Applying this method on the MMLU dataset with CaMVo ($\delta=0.96$, $k_{\min}=1$) yields an accuracy of 86.96\% at a cost of 3.07, exceeding the target accuracy of 84.65\%. For comparison, the original CaMVo results are 87.98\% accuracy at 4.74 cost ($\delta=0.96$), and 86.82\% accuracy at 2.76 cost ($\delta=0.95$). On the IMDB dataset, the simulation with $\delta=0.99$ and $k_{\min}=1$ produces 95.09\% accuracy at a cost of 0.83, surpassing the target accuracy of 94.66\%. The original CaMVo results are 95.10\% accuracy at 1.09 cost ($\delta=0.99$), and 94.69\% accuracy at 0.34 cost ($\delta=0.985$).
>
> These results indicate that incorporating the copula-based simulation does not significantly improve performance on MMLU and yields a modest cost improvement on IMDB, where LLM outputs exhibit higher correlation. We attribute this to two factors: (1) LinUCB’s use of input embedding vectors to predict LLM confidences already implicitly captures correlations among LLM outputs conditioned on the input context, and (2) estimation errors in the correlation matrix reduce the potential gains from explicitly modeling correlations.
>
> We plan to extend this study across all tested $\delta$ and $k_{\min}$ values in our simulations and include these results in the final version of the paper.
>
> Additionally, Lemma 2.1 can be extended to remove the conditional-independence assumption using Bahadur’s model (R.R. Bahadur, “On classification based on responses to n dichotomous items,” 1961) characterizes the joint distribution of $n$ binary variables using their marginal probabilities $p_i$, second-order dependence terms known as Bahadur parameters, and higher-order interaction terms. This framework allows the computation of majority vote confidence by summing the probabilities of all outcomes where the majority label is correct. However, the lack of a closed-form solution and the rapid growth in computational complexity with increasing $n$ limit its practical applicability. We will include a note on this in the discussion of Lemma 2.1 in the final version of the paper.
>
>
> ## Responses to Weaknesses:
>
> **W1:** The NP-hard nature of subset selection is well known in the combinatorial bandits literature (e.g., CUCB, Combinatorial Thompson Sampling), and many algorithms assume access to an oracle (either exact or approximate) that solves the offline combinatorial optimization step.
>
> An exact oracle, which evaluates all possible subsets, has a time complexity of $O(2^K)$ per round. However, for small $K$ (i.e., $K \leq 10$), the number of subsets remains tractable. In our simulations, $K=7$, and we did not observe any meaningful impact on runtime.
>
> For larger $K$, an approximate oracle can be employed to reduce computation at the cost of approximation. In our setting, one such approach is to use Monte Carlo simulations to generate $m$ realizations for each LLM. Given these samples, the confidence of a subset of size $K$ can be estimated in $O(mK)$ time. As we discussed in our response to Q2, this approach can also incorporate LLM correlations using a given correlation matrix, however time complexity will increase to $O(mK^2 + K^3)$. To efficiently search a low-cost subset that satisfies the confidence threshold, a greedy approximation can be used. Starting with all LLMs in the subset, at each step we can remove the highest cost LLM that does not reduce the confidence of the subset below the threshold. This will continue until no more LLMs can be removed. In this method, each iteration will have $O(K)$ comparisons, and in total there will be $O(K)$ iterations. As confidence estimation for one subset is $O(mK)$, in total the runtime will be $O(mK^3)$ for this method. We will add this discussion to the final version of the paper.
>
> **W2:** Please refer to our response Q2.
>
> ## Responses to Limitations:
>
> **L1:** Please refer to our response Q2.

---

> > ### Comment · Reviewer_ZiN8 · 2025-08-08
> >
> > Thank you for the rebuttal. My concerns regarding the independence assumption and NP-hard complexity have been addressed. The proposed solutions are practical and sufficient. I recommend acceptance.

---

### Official Review · Reviewer_fBz7 · 2025-07-03

**Clarity:** 3
**Significance:** 3
**Originality:** 2
**Rating:** 5
**Confidence:** 3

**Summary:**

The paper proposes a method to approximate the majority voting of an ensemble of LLMs in an online setting, using a subset of the ensemble. They aim to achieve comparable accuracy to the full-ensemble at much lower costs. The method is supported by experiments showing that their method is close to the Pareto-optimal frontier of cost and accuracy.

**Questions:**

- As mentioned in the weaknesses, the choice of priors for the label correctness and regularization is not clear. Could you give a justification? How would it compare to different choices?

**Ethical Concerns:**

["NO or VERY MINOR ethics concerns only"]

**Final Justification:**

The authors addressed the issues I mentioned before, that is, revising the terminology to make it more clear, explaining the modeling choices and adding a sensitivity study to evaluate the impact of correlated LLM outputs (preliminary in the rebuttal and promise of extension in the final version).

**Limitations:**

Yes.

**Paper Formatting Concerns:**

No concerns.

**Quality:**

3

**Strengths And Weaknesses:**

#### Strengths
- The paper is well-written and easy to follow for the most part.
- I appreciated the in-depth experiments, especially the cost accuracy trade off plots in Figure 1 and 2.

#### Weaknesses
- The language in section 3 is imprecise and might be misleading. For instance, $y_t$ is defined the predicted label by aggregating the votes of the LLM ensemble in a weighted way, see eq. (1). As the authors say that the true label is not available, they replace it with the *ensemble prediction* and define *empirical accuracy* as agreeing with the *ensemble prediction*. This has nothing to do with *accuracy* as we know it, i.e., agreeing with the *true label*. Note that the majority vote (based on weights proportional to accuracy) and their so called baseline method (based on weights proportional to agreeing with the previous ensemble prediction) perform very similar in experiments. However, it is unclear whether (1) the weights are actually similar, (2) both weights are unstable but the aggregation is not sensitive or (3) weights are both close to uniform. Moreover, if the weights were close, this might be a results of the ensemble accuracy being rather high, i.e. above 85%, and would not hold in a more general case.
- The authors model the the label correctness as a beta distributed random variable, but give no justification for this choice. I only found that the parameters of the shape beta distribution are estimated by MM oder MLE in the supplements.
- The authors propose a regularization based on Laplace smoothing to avoid overconfident weights and bias aggregation, again without any reasoning.
- Table 3 is a bit misleading. As the best single model accuracy is 85.92% by o3-mini in Table 2, there is no reason to apply their method to ensemble a subset of LLMs to attain a much lower target accuracy at higher costs. This is the case in all but the first few rows. Similarly, the individual models are missing in the cost accuracy trade off plots in Figure 1 and 2, or are at least not clearly indicated.
- Although the authors mention that their method is based on the independence of LLM predictions, they do not conduct a proper sensitivity study on how vulnerable their method is.

---

> ### Author Rebuttal · Authors · 2025-07-31
>
> Dear Reviewer,
>
> Thank you for the comments. Our responses to the reviewer's comments are below.
>
> ## Responses to Weaknesses:
>
> **W1:** In our paper, we consider a problem setting where a user wished to label a dataset. Without our method, the default approach would be majority voting. Our primary motivation is to reduce the labeling cost by allowing a trade-off with accuracy. Accordingly, the accuracy we define is relative to the accuracy of majority voting, which serves as the baseline for comparison. We acknowledge that this distinction could be more clearly articulated. In the final version, we will revise the terminology to avoid ambiguity. For instance, we will "accuracy" to "relative accuracy" and "target accuracy" to "target relative accuracy". Note that the accuracy values reported in our simulation results correspond to comparisons between the generated labels and the true labels, so the use of "Accuracy" in those contexts remains appropriate.
>
> Regarding the LLM accuracies, the individual model accuracies (used as weights in majority voting) on the MMLU dataset are
> [0.680,0.848,0.817,0.835,0.856,0.859,0.640], and the final weights learned by the baseline method are [0.728,0.905,0.875,0.893,0.901,0.911,0.689]. As expected, these weights are higher than the original accuracies, since under the baseline method, any LLM that agrees with the majority vote is treated as having produced a correct label—even in cases where all LLMs are jointly incorrect. This inflates the estimated accuracy. Nonetheless, the baseline performs comparably to majority voting because for the majority voting output to be correct, the weights of the correct labels need to exceed the weights of incorrect labels. Even though the weights are not exactly correct, the baseline method learns the correct ordering of the weights (if LLMs are sorted by weight both set of weights produce the same result). With this, the weights of the correct labels can still exceed the weights of incorrect labels, and this helps the baseline method achieve similar results as majority voting. This explains why the baseline method achieves results close to majority voting.
>
>
> **W2:** Using a Beta distribution is a standard Bayesian approach for modeling binary outcomes, as it is the conjugate prior to the Bernoulli likelihood. This allows for efficient incremental posterior updates as new binary observations (correct/incorrect labels) are collected. While maximum likelihood estimation (MLE) is the formal method for updating the shape parameters of the Beta distribution based on observed data, it has no closed-form solution. In practice, the method of moments is commonly used as a computationally efficient alternative to approximate the parameter updates.
>
>
> **W3:** Since no labeled data is available at the start, early accuracy estimates of the LLMs can be noisy and prone to overfitting. Because subset selection is based on these estimated confidences, early estimation errors can bias the selection process and lead to compounding errors over time. To mitigate this, we use regularization to stabilize the accuracy estimates. We choose Laplace smoothing as the regularizer as Laplace smoothing corresponds to using a uniform prior on the Beta estimator, and hence combines well with the Beta estimator. We use a log(t) term with the Laplace smoothing as compared to a constant term, as the log(t) term does not decay as quickly as the constant term and prevents overfitting in the long run.
>
> We will include the reasoning behind our design choices in the final version of the paper.
>
>
>
>
> **W4:** While we use the true labels to compute accuracy in our simulations, these labels are not available to the user in the actual problem setting. As a result, the user does not know a priori which LLM performs best, or whether any individual model outperforms majority voting. Crucially, if only a single LLM is queried in each round, its output will be treated as the true label. Since it will always agree with itself, this prevents any meaningful estimation of the true accuracy of the LLM. Hence, to reliably estimate individual LLM accuracies over time, it is necessary to query a subset of LLMs in each round, which incurs a cost. This is a fundamental aspect of the exploration-exploitation trade-off in our framework.
>
> Regarding the cost-accuracy trade-off plots, although individual models are already included in the plots, we agree that their presentation can be improved. In the final version, we will use a distinct color to clearly differentiate individual LLMs for better visual clarity.
>
>
> **W5:** We have conducted the following sensitivity study to evaluate the impact of correlated LLM outputs. Using the true accuracies of the LLMs, we generate synthetic labels via a Gaussian copula approach. This method preserves the original accuracies while introducing correlations among the LLMs as specified by a correlation matrix. In this matrix, the diagonal entries representing self-correlation are set to 1, and all off-diagonal entries are set to a parameter $C_i$ that represents the pairwise correlation. We run experiments using these synthetic labels, sweeping $C_i$ across a range of values to analyze how varying correlation strengths affect performance.
>
> The results of this experiment for the MMLU dataset are given below .
> |        | $C_i=0$   | $C_i=0.1$   | $C_i=0.2$   | $C_i=0.3$   | $C_i=0.4$   | $C_i=0.5$  | $C_i=0.6$   | $C_i=0.7$   | $C_i=0.8$   |
> |--------|--------|--------|--------|--------|--------|--------|--------|--------|--------|
> | CaMVo ($\delta=0.96, k_{min}=1$) | 96.67  | 94.84  | 93.24  | 91.72  | 89.53  | 88.19  | 87.01  | 84.02  | 78.96  |
> | Maj vote  | 96.66  | 94.15  | 91.46  | 89.88  | 87.67  | 86.35  | 84.53  | 83.59  | 82.49  |
> | Target acc.  | 92.79  | 90.38  | 87.80  | 86.29  | 84.16  | 82.90  | 81.15  | 80.25  | 79.19  |
>
> As can be seen from the results, although the performance of CaMVo declines as the correlation among LLMs increases, the performance of majority voting also deteriorates similarly. Importantly, CaMVo is able to maintain the target accuracy across all tested correlation levels except at the very high correlation coefficient of $C_i=0.8$, verifying the utility of CaMVo even in correlated regimes.
>
> Currently, we have used a single uniform value for all off-diagonal entries in the cross-correlation matrix. In the final version of the paper, we plan to extend this sensitivity study by exploring varying cross-correlation values across different matrix elements. Additionally, we will extend the experiments to the IMDB dataset and conduct experiments with a broader range of $\delta$ values to provide a more comprehensive analysis.
>
>
>
> ## Responses to Questions:
>
> We have addressed the reviewer’s questions as part of our responses to the weaknesses.

---

> > ### Comment · Reviewer_fBz7 · 2025-08-03
> >
> > Thank you for deeply engaging with the points I raised. I appreciated in particular the explanations to your modeling choices and sensitivity study to evaluate the impact of correlated LLM outputs.

---

> > ### Author Response · Authors · 2025-08-06
> > **Additional experimental results for correlation sensitivity study with synthetic labels**
> >
> > Dear Reviewer,
> >
> > We would like to follow up on our rebuttal regarding experiments with synthetic labels as we have now completed the experimental run on the IMDB dataset. As a reminder, these experiments use synthetic labels designed to match the accuracy of the original LLM responses, and sweep $C_i$ across a range of values to analyze how varying correlation strengths affect performance. The results on the IMDB dataset are presented below:
> >
> > |        | $C_i=0$   | $C_i=0.1$   | $C_i=0.2$   | $C_i=0.3$   | $C_i=0.4$   | $C_i=0.5$  | $C_i=0.6$   | $C_i=0.7$   | $C_i=0.8$   |
> > |--------|--------|--------|--------|--------|--------|--------|--------|--------|--------|
> > | CaMVo ($\delta=0.98, k_{min}=1$) | 98.70 | 98.31 | 97.89 | 97.36 | 96.66 | 96.09 | 95.52 | 95.28 | 94.70 |
> > | Maj. vote | 99.96 | 99.83 | 99.53 | 99.17 | 98.66 | 98.08 | 97.37 | 96.82 | 95.96 |
> > | Target acc. | 97.96 | 97.83 | 97.54 | 97.19 | 96.69 | 96.12 | 95.42 | 94.88 | 94.04 |
> >
> > As observed, the results align closely with those from the MMLU dataset. Both CaMVo and majority voting exhibit performance degradation as correlation coefficient increases. Importantly, CaMVo consistently meets or exceeds the target accuracy across all correlation levels, except at $C_i=0.4$ and $C_i=0.5$, where it falls slightly below. We attribute this small deviation to the limited number of trials currently used (10 runs per setting). We plan to increase the number of trials in the final version to improve the stability of the results, and also to conduct more experiments with a broader range of $\delta$ values.

---

### Note · Authors · 2025-08-14

Dear Area Chairs,

We would like to express our gratitude for the time, expertise, and attention of the reviewers in reviewing our manuscript. We appreciate that the reviewers acknowledge the strengths of our work including its practical utility and "strong theoretical foundation." We have addressed all the issues raised by the reviewers, and the reviewers have responded positively to our rebuttals with three reviewers indicating they will recommend acceptance and another appreciating our rebuttal responses. We believe the review process has enriched the quality and rigor of our work, and our paper now constitutes a valuable contribution in LLM-based online dataset annotation. For your reference, below is a summary of the key improvements we have made during the rebuttal period, and our planned improvements.

Key improvements:\
**Sensitivity to correlation study:** In response to the concerns of Reviewers fBz7 and Ld4S, we analyzed CaMVo’s performance using Gaussian copula–generated labels with varying LLM correlations. While accuracy decreases for both CaMVo and majority voting as correlation rises, CaMVo maintains target accuracy except at very high correlation, confirming robustness in correlated settings.\
**Modeling correlations between LLMs:** Reviewers ZiN8, Ld4S, and U74S noted that Lemma 2.1’s conditional independence assumption may be unrealistic. In our rebuttal, we showed it can be relaxed via Bahadur’s model, which captures pairwise and higher-order dependencies, but its lack of a closed form and exponential complexity make it impractical. We therefore proposed a Gaussian copula–based Monte Carlo approach to generate correlated binary outcomes from LLM accuracies and correlations, enabling empirical estimation of majority vote confidence. On MMLU and IMDB, this produced minimal accuracy gains, likely due to noisy correlation estimates and LinUCB’s implicit correlation handling.\
**NP-hard nature of subset selection:** We have provided a practical approximation-based subset selection method that runs in polynomial time based on the comments of reviewer ZiN8.

Planned improvements:\
**Empirical validation:** We will add experiments with at least one more real-world dataset.\
**Sensitivity to correlation study:** We will extend the study to all experimental parameter settings and to correlation matrices with more varied entries.\
**Modeling correlations between LLMs:** We will extend the study of the new method across all parameter settings.

---

### Decision · Program_Chairs · 2025-09-17

**Decision:**

Accept (spotlight)

**Comment:**

The paper propose CaMVo, a cost-aware framework for dataset annotation with LLMs. The idea is to select adaptively a subset of models using contextual bandit with Bayesian estimation, in this way saving cost but keeping accuracy. This is a relevant and timely problem because LLM annotation is becoming very expensive, and the approach is simple but novel.

Experiments are convincing: results are close or better than majority voting but with much lower cost. The method is well motivated and ablations are strong. Some concerns on assumptions were raised but authors reply was reasonable. Overall I think the paper is solid and useful for the community, so I recommend accept.